# Adversarial Self-Training Improves Robustness and Generalization for Gradual Domain Adaptation

**Lianghe Shi**     **Weiwei Liu**[*]
School of Computer Science, Wuhan University
National Engineering Research Center for Multimedia Software, Wuhan University
Institute of Artificial Intelligence, Wuhan University
Hubei Key Laboratory of Multimedia and Network Communication Engineering, Wuhan University

## Abstract

Gradual Domain Adaptation (GDA), in which the learner is provided with additional intermediate domains, has been theoretically and empirically studied in many contexts. Despite its vital role in security-critical scenarios, the adversarial robustness of the GDA model remains unexplored. In this paper, we adopt the effective gradual self-training method and replace vanilla self-training with adversarial self-training (AST). AST first predicts labels on the unlabeled data and then adversarially trains the model on the pseudo-labeled distribution. Intriguingly, we find that gradual AST improves not only adversarial accuracy but also clean accuracy on the target domain. We reveal that this is because adversarial training (AT) performs better than standard training when the pseudo-labels contain a portion of incorrect labels. Accordingly, we first present the generalization error bounds for gradual AST in a multiclass classification setting. We then use the optimal value of the Subset Sum Problem to bridge the standard error on a real distribution and the adversarial error on a pseudo-labeled distribution. The result indicates that AT may obtain a tighter bound than standard training on data with incorrect pseudo-labels. We further present an example of a conditional Gaussian distribution to provide more insights into why gradual AST can improve the clean accuracy for GDA.

## 1  Introduction

The key assumption of classical machine learning—that training and test data come from the same distribution—may not always hold in many real-world applications [15]. A data distribution typically evolves due to changes in conditions: for example, changing weather in vehicle identification [14], sensor aging in sensor measurement [43], the evolution of road conditions in self-driving [6], etc. To address this problem, Unsupervised Domain Adaptation (UDA) has been developed to train a model that performs well on an unlabeled target domain by leveraging labeled data from a similar yet distinct source domain.

Various works in UDA [5, 53, 34] theoretically demonstrate that the generalization error can be controlled by the domain discrepancy. Hence, the domain shift between the source domain and the target domain is expected to be small [15]. However, in some applications, the domain shift is substantial, leading to a sharp drop in the performance of the UDA method [26]. Furthermore, since changes in real-world data are more often gradual than abrupt [16], there are many intermediate domains between the source domain and the target domain. To get a better solution to the gradually shifting data, some recent works have focused on the Gradual Domain Adaptation (GDA) problem, where the learner is additionally provided with unlabeled intermediate domains. The large gap between the source and target domains is then divided up into multiple small shifts between the

---

[*]Correspondence to: Weiwei Liu <liuweiwei863@gmail.com>.

37th Conference on Neural Information Processing Systems (NeurIPS 2023).

intermediate domains. Recently, the celebrated work [26] proposes gradual self-training, which iteratively applies the self-training method to adapt the model along the intermediate domains. Empirically, the gradual self-training method greatly improves the target domain's accuracy compared to the traditional direct domain adaptation. After that, remarkable theoretical [45] and algorithmic [10, 1] advances have been achieved in GDA. However, the adversarial robustness of the GDA model remains unexplored, despite its vital role in security-critical scenarios. Adversarial robustness refers to the invariance of a model to small perturbations of its input [39], while adversarial accuracy refers to a model's prediction accuracy on adversarial examples generated by an attacker. Numerous works [33, 41, 49] have shown that very small perturbations, even those imperceptible to humans, can successfully deceive deep neural network (DNN) models, resulting in erroneous predictions.

One popular and effective method of improving robustness is Adversarial Training (AT), which adds adversarial examples to the training data. Due to the lack of labels in the target domain, it is difficult to directly generate adversarial examples. Accordingly, this paper replaces self-training with Adversarial Self-Training (AST), which first generates pseudo-labels on the unlabeled data and then adversarially trains the model based on these pseudo-labeled data. We conduct comprehensive experiments to validate the effectiveness of gradual AST in Section 3. Compared to the gradual self-training method, the proposed gradual AST method delivers a great improvement in adversarial accuracy, from $6.00\%$ to $90.44\%$ on the Rotating MNIST dataset. More interestingly, we find that the clean accuracy of gradual AST on the target domain also increases from $90.06\%$ to $97.15\%$. This is a surprising result, since many prior works [52, 38] demonstrate that AT may hurt generalization; in other words, there is a trade-off between clean accuracy and adversarial accuracy. We then empirically investigate the reason why AST improves clean accuracy in GDA and find that the adversarially trained model has better clean accuracy than the standardly trained model when the generated pseudo-labels contain a proportion of incorrect labels.

In Section 4, we present novel generalization error bounds for the proposed gradual AST. The results show that if we have a model $f$ with low adversarial margin loss on the source domain and the distribution shifts slightly, gradual AST can generate a new model $f'$ that is also adversarially robust on the target domain. To explain why gradual AST achieves better clean accuracy than vanilla gradual self-training, we use the optimal value of the Subset Sum Problem [3] to bridge the standard error on a real distribution and the adversarial error on a pseudo-labeled distribution. The result shows that AT may yield a tighter generalization guarantee than standard training on pseudo-labeled training data. We further provide an example of a conditional Gaussian distribution to illustrate why AT outperforms standard training when a proportion of the labels in the training data are incorrect.

We summarize our contributions as below:

- We are the first to apply the AST method in GDA. The proposed gradual AST method can not only improve the adversarial robustness but also the clean accuracy for GDA, which is an appealing and nontrivial result, considering that many prior works demonstrate that adversarial training may hurt generalization.

- We empirically explore the reason for the improvements in clean accuracy and find that adversarial training performs better than standard training when the pseudo-labeled training set contains a proportion of incorrect labels.

- From the theoretical perspective, we provide the generalization error bounds for gradual AST, which explain why the trained model is adversarially robust on the target domain. We provide an error bound and a toy example of Gaussian distribution to provide some intuitions for the improvements in clean accuracy.

## 2 Preliminaries

### 2.1 Multiclass Classification Learning Framework

Let $\mathcal{Z} = \mathcal{X} \times \mathcal{Y}$ be a measurable instance space, where $\mathcal{X}$ and $\mathcal{Y}$ denote the input space and label space, respectively. The input space $\mathcal{X}$ is a subset of a $d$-dimensional space, $\mathcal{X} \subseteq \mathbb{R}^d$. In this work, we focus on multiclass classification, and the label space $\mathcal{Y}$ is $\{1, \ldots, k\}$, where $k$ is the number of classes. Following the notations used in [5], a domain is envisioned as a tuple $\langle P, h_P \rangle$, consisting of a distribution $P$ on input space $\mathcal{X}$ and a labeling function $h_P : \mathcal{X} \to \mathcal{Y}$. In practice, the true distribution $P$ is unknown to the learner, which has access only to the training data $S$ drawn

independent and identically distributed (i.i.d.) according to the true distribution $P$. We use $\widehat{P}$ to denote the empirical distribution of $P$ according to the training data $S$. Moreover, following the notations used in [35], we consider the class $\mathcal{F} = \{f : \mathcal{X} \to \mathbb{R}^k\}$ of scoring functions $f$. We use $f_y(x)$ to indicate the output of $f$ associated with the data point $x$ on the $y$-th dimension; the output indicates the confidence of the prediction. The label with the largest score is the predicted label of $x$. We use $\mathcal{H}_{\mathcal{F}} = \{h_f(\cdot) = \arg\max_{y \in \mathcal{Y}} f_y(\cdot) : f \in \mathcal{F}\}$ to denote the labeling function class induced by $\mathcal{F}$. The expected risk and the empirical risk of a classifier $h \in \mathcal{H}$ with respect to a labeling function $h'$ on distribution $P$ are defined as $R_P(h, h') \triangleq \mathbb{E}_{x \sim P} \mathbb{1}[h(x) \neq h'(x)]$ and $R_{\widehat{P}}(h, h') \triangleq \mathbb{E}_{x \sim \widehat{P}} \mathbb{1}[h(x) \neq h'(x)]$, respectively, where $\mathbb{1}$ is the indicator function. We use $R_P(h)$ and $R_{\widehat{P}}(h)$ to abbreviate $R_P(h, h_P)$ and $R_{\widehat{P}}(h, h_P)$ respectively. Furthermore, the 0-1 loss is non-differentiable and cannot be minimized directly. Thus, a margin theory for multiclass classification was developed by [25], that replaces the 0-1 loss with the margin loss. The margin of a scoring function $f \in \mathcal{F}$ for labeled data $(x, y)$ is defined as $\rho_f(x, y) \triangleq f_y(x) - \max_{y' \neq y} f_{y'}(x)$. And the expected margin loss of a scoring function $f \in \mathcal{F}$ with respect to another scoring function $f' \in \mathcal{F}$ on distribution $P$ is defined as $R_P^{(\rho)}(f, f') \triangleq \mathbb{E}_{x \sim P} \Theta_\rho \circ \rho_f(x, h_{f'}(x))$, where $\Theta_\rho(m) = \min\{1, \max\{0, 1 - m/\rho\}\}$ is the ramp loss. Similarly, we use $R_P^{(\rho)}(f)$ and $R_{\widehat{P}}^{(\rho)}(f)$ to abbreviate $R_P^{(\rho)}(f, f_P)$ and $R_{\widehat{P}}^{(\rho)}(f, f_P)$ respectively. Note that $R_P(h_f, h_{f'}) \leq R_P^{(\rho)}(f, f')$, since the 0-1 loss is upper bounded by the margin loss.

## 2.2 Gradual Domain Adaptation

Under the standard GDA settings [26, 45], the learner is sequentially trained on $T + 1$ domains with gradual shifts. The corresponding data distributions are $P_0, P_1, \ldots, P_T$, where $P_0$ is the distribution of the source domain, $P_T$ is the distribution of the target domain, and $P_1, \ldots, P_{T-1}$ are the distributions of the intermediate domains. For simplicity, we assume that the number of data points in each domain is the same: namely, each domain $t$ has a set of $n$ data drawn i.i.d. from $P_t$, denoted as $S_t$. Recently, [26] proposes a gradual self-training method for GDA, which successively applies self-training to each of the intermediate domains. The self-training method adapts a pre-trained model trained on the previous domain to the current domain using unlabeled data drawn from the current distribution. Specifically, given a pre-trained model $f$ and an unlabeled data set $S$, the model first predicts the labels of the unlabeled data, then trains $f$ with empirical risk minimization (ERM) over these pseudo-labeled data, i.e.,

$$f' = ST(f, S) = \arg\min_{f' \in \mathcal{F}} R_{\widehat{P}}^{(\rho)}(f', f) = \arg\min_{f' \in \mathcal{F}} \frac{1}{n} \sum_{x_i \in S} \Theta_\rho \circ \rho_{f'}(x_i, h_f(x_i)).$$

In gradual self-training, the model is first pre-trained on the labeled source domain and then successively self-trained on the unlabeled dataset of each of the intermediate domains, i.e.,

$$f_t = ST(f_{t-1}, S_t), t \in \{1, \ldots, T\}.$$

Since the domain shifts between the consecutive domains are assumed to be small [10, 26, 45], the accuracy of the pseudo-labels is expected to be high at each step. The gradual self-training method aims to output a final trained classifier $h_{f_T}$ with low expected risk in the target domain.

## 2.3 Adversarial Self-Training

Given a classifier $h_f$ and a data point $(x, y)$, we can generate the corresponding adversarial example $(x^{adv}, y)$ by adversarially perturbing $x$ in a small neighborhood $B_\epsilon(x)$ of $x$, as follows: $x^{adv} = \arg\max_{x' \in B_\epsilon(x)} \Theta_\rho \circ \rho_f(x', y)$. In this paper, we focus on the $\ell_q$ adversarial perturbation $B_\epsilon(x) \triangleq \{x' \in \mathcal{X} : \|x' - x\|_q \leq \epsilon, q \geq 1\}$, which is referred to as $\ell_q$-attack in the AT context and has been widely studied in existing works. Given a vector $x \in \mathbb{R}^d$, we define the $\ell_q$-norm of $x$ as $\|x\|_q \triangleq \left(\sum_{i=1}^d |x_{(i)}|^q\right)^{1/q}$, for $q \in [1, \infty)$, where $x_{(i)}$ is the $i$-th dimension of $x$; for $q = \infty$, we define $\|x\|_\infty \triangleq \max_{1 \leq i \leq d} |x_{(i)}|$. Similar to the definition of the standard risk $R_P(h, h')$, we

define the adversarial risk on distribution $P$ for any two classifiers $h, h'$ as follows: $\widetilde{R}_P(h, h') \triangleq \underset{x \sim P}{\mathbb{E}} \underset{\|\delta\|_q \leq \epsilon}{\max} \mathbb{1}[h(x + \delta) \neq h'(x)]$. We also define the expected adversarial margin loss for any two scoring functions $f, f'$ on distribution $P$ as follows:

$$\widetilde{R}_P^{(\rho)}(f, f') \triangleq \underset{x \sim P}{\mathbb{E}} \underset{\|\delta\|_q \leq \epsilon}{\max} \Theta_\rho \circ \rho_f(x + \delta, h_{f'}(x)). \tag{1}$$

Although the expected adversarial margin loss is asymmetrical, it satisfies the triangle inequality (see proof in Appendix A.1). The empirical adversarial risk and the empirical adversarial margin loss can be defined similarly. We use $\widetilde{R}_P^{(\rho)}(f)$ and $\widetilde{R}_{\widehat{P}}^{(\rho)}(f)$ to abbreviate $\widetilde{R}_P^{(\rho)}(f, f_P)$ and $\widetilde{R}_{\widehat{P}}^{(\rho)}(f, f_P)$ respectively. To train a robust model with resistance to the adversarial examples, AT aims to find a classifier $h_f$ that yields the minimal adversarial margin loss, i.e., $\underset{f \in \mathcal{F}}{\arg\min} \widetilde{R}_{\widehat{P}}^{(\rho)}(f)$. However, we cannot adversarially train the model directly on the unlabeled target domain because the conventional AT method requires labeled data. Instead, we adapt the self-training method to the adversarial self-training (AST) method by generating adversarial examples over the pseudo-labels, i.e.,

$$f' = AST(f, S) = \underset{f' \in \mathcal{F}}{\arg\min} \widetilde{R}_{\widehat{P}}^{(\rho)}(f', f) = \underset{f' \in \mathcal{F}}{\arg\min} \frac{1}{n} \sum_{x_i \in S} \underset{\|\delta_i\|_q \leq \epsilon}{\max} \Theta_\rho \circ \rho_{f'}(x_i + \delta_i, h_f(x_i)).$$

## 3 Empirical Exploration

In this section, we propose a gradual AST method to improve adversarial robustness. Surprisingly, we find that the gradual AST improves not only adversarial robustness but also clean accuracy. Our code is available at https://github.com/whustone007/AST_GDA.

**Datasets.** We run experiments on two datasets that are widely used in GDA [26, 45]. **Rotating MNIST** is a semi-synthetic dataset generated by rotating each MNIST image at an angle between 0 and 60 degrees. The 50,000 training set images are divided into three parts: a source domain of 5000 images (0-5 degrees), 21 intermediate domains of 42,000 images (5-60 degrees), and a set of validation data. The rotating degree gradually increases along the domain sequence. The 10,000 test set images are rotated by 55-60 degrees, representing the target domain. **Portraits** [18] is a real dataset consisting portraits of American high school students across a century. The model aims to classify gender. The dataset is split into a source domain of the first 2000 images, seven intermediate domains of the next 14,000 images, and a target domain of the next 2000 images.

**Methods.** The standard gradual self-training method [26] successively adapts the model to the next domain via self-training. To improve the adversarial robustness, we replace the vanilla self-training with AST. We implement multiple gradual AST methods with different starting domains $\tau \in \{0, \ldots, T + 1\}$ where we begin to use AST, i.e.,

$$f_t = \begin{cases} ST(f_{t-1}, S_t), & 1 \leq t \leq \tau - 1 \\ AST(f_{t-1}, S_t), & \tau \leq t \leq T \end{cases}$$

In particular, if $\tau = 0$, then the model is adversarially trained throughout; if $\tau = T + 1$, then the model is standardly trained throughout. Note that $T = 21$ in Rotating MNIST and $T = 7$ in Portraits.

**Implementation Details.** We implement our methods using PyTorch [37] on two Nvidia GeForce RTX 3090 Ti GPUs. Following [26], we use a 3-layer convolutional network with dropout (0.5) and BatchNorm on the last layer. We use mini-batch stochastic gradient descent (SGD) with momentum 0.9 and weight decay 0.001. The batch size is 32 and the learning rate is 0.001. We train the model for 40 epochs in each domain. We set the radius of the bounded perturbation to be 0.1 [51] for Rotating MNIST and 0.031 for Portraits. Following [33], we use PGD-20 with a single step size of 0.01 as the adversarial attacker.

### 3.1 Results

The results of the different methods on Rotating MNIST are shown in Figure 1(a). The numbers in the abscissa represent different starting domains $\tau$ of various methods. From Figure 1(a), we can see that the vanilla gradual self-training method ($\tau = 22$) achieves a clean accuracy of 90.06% and adversarial accuracy of 6.00% on the target domain. Notably, the gradual AST method that uses AST

throughout ($\tau = 0$) improves the adversarial accuracy to $83.54\%$ ($+77.54\%$). More interestingly, we find that the clean accuracy is also slightly improved by $0.53\%$. Furthermore, the results of varying the starting domains $\tau$ indicate that the performance will be further improved if we use the vanilla self-training method for the first few domains and then use AST from an intermediate domain. We utilize the validation set of the target domain to choose the optimal starting time $\tau$. Experimentally, the best choice of starting time for Rotating MNIST is $\tau = 9$, which achieves clean accuracy of $96.66\%$ ($+6.6\%$) and adversarial accuracy of $88.86\%$ ($+82.86\%$). We find that AST not only improves adversarial robustness but also clean accuracy with a large step. The results of the different methods on Portraits are shown in Figure 1(b). We find similar results to those on Rotating MNIST: the vanilla gradual self-training method ($\tau = 22$) achieves a clean accuracy of $82.03\%$ and adversarial accuracy of $40.23\%$, while the gradual AST method ($\tau = 0$) improves the clean accuracy by $2.73\%$ and the adversarial accuracy by $37.40\%$. We provide more experimental results of different $\epsilon$, domain numbers, and neural networks in Appendix B.

### 3.2 Why Does Gradual AST Improve Clean Accuracy in GDA?

It is widely believed that AT would hurt generalization [52, 38] and decrease clean accuracy; thus it is surprising that the proposed gradual AST method improves the clean accuracy of the trained model in the GDA setting. In this section, we speculate on the reasons for this. Recently, some works [39, 42] have revealed that an adversarially pre-trained model tends to improve the target model's accuracy in transfer learning. We accordingly investigate whether the adversarially trained model has better clean accuracy on the next domain and thus generates more accurate pseudo-labels. However, by comparing the gradual AST methods with $\tau = 0$ and $\tau = 1$ on Rotating MNIST, we find that the model standardly trained on the source domain achieves a clean accuracy of $97.30\%$ on the next domain, while the model adversarially trained on the source domain achieves a clean accuracy of $96.95\%$ on the next domain. On the other hand, given that the pseudo-labels contain a small portion of incorrect labels, we wonder whether the adversarially trained model has better clean performance than the standardly trained model if trained on such a noisy training set. Since we set a fixed random seed for all gradual AST methods with varying starting domains, given a $\tau_0 \in \{0, \ldots, T\}$, the gradual AST method with $\tau = \tau_0$ and that with $\tau = \tau_0 + 1$ have the same training processes before domain $\tau_0$ and the same pseudo-label set on domain $\tau_0$. The difference between the gradual AST method with $\tau = \tau_0$ and that with $\tau = \tau_0 + 1$ is whether we adversarially train the model on the domain $\tau_0$. We compare the clean accuracy of these two methods on domain $\tau_0$ in Figure 1(c). As we can observe from the figure, the adversarially trained model ($\tau = \tau_0$) has higher clean accuracy on domain $\tau_0$ than the standardly trained model ($\tau = \tau_0 + 1$). Note that there are two exceptions in which AT hurts clean performance when the accuracy of pseudo-labels is very high ($\tau_0 = 0, 1$), which explains why using AST from an intermediate domain is better than using AST throughout. Besides, we find that the adversarially trained model predicts more accurate pseudo-labels in the next domain than the standardly trained model when the current training set contains a proportion of incorrect labels. We then conclude that AT benefits the clean accuracy of GDA from two perspectives:

- For the intermediate domains, AT improves the accuracy of the pseudo-labels in the next domain. So the learner can fine-tune the model on a more accurate dataset.

- For the target domain, when the pseudo-labels contain a proportion of incorrect labels, the adversarially trained model has higher clean accuracy on the target domain than the standardly trained model.

### 3.3 Sensitivity of Filtration Ratio $\zeta$

Following the standard experimental strategy [26, 45], for each step of self-training, we filter out a portion of images for which the model's prediction is least confident. In Figure 1, the ratio $\zeta$ is set to $0.1$ as in the work of [26]. We further conduct more experiments with varying $\zeta$. With smaller $\zeta$, more data will be included for the next domain's training, but the accuracy of the pseudo-labels will decrease. Due to the page limit, we attach the detailed results of methods with different values of $\tau$ and $\zeta$ in Appendix B. We find that when $\zeta$ decreases, the performance of the standard gradual self-training also decreases. However, the gradual AST method prefers smaller $\zeta$ even though more incorrect pseudo-labels are included in the training set. For example, when we set $\zeta = 0.05$, the gradual AST with the optimal starting domain $\tau = 9$ (chosen using the validation set) achieves clean accuracy of $97.15\%$ ($+7.09\%$) and adversarial accuracy of $90.44\%$ ($+84.44\%$) on Rotating MNIST.

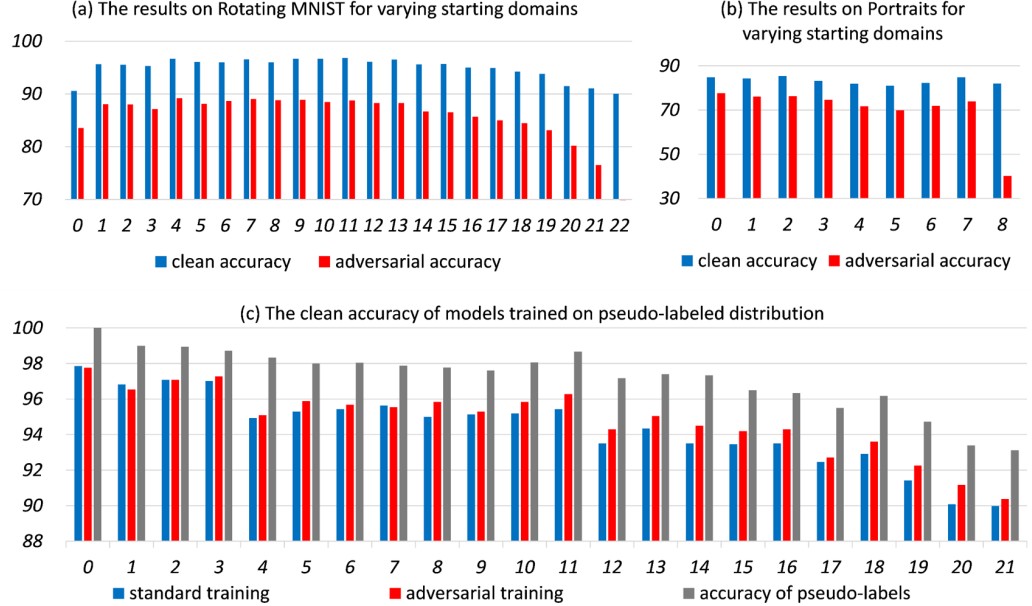

Figure 1: (a) shows the results of gradual AST methods with varying starting domains on Rotating MNIST. The abscissa indicates different methods with varying starting domains $\tau$. The blue and red bars represent the clean accuracy and adversarial accuracy respectively of the trained model on the target domain. (b) shows the result of gradual AST methods with varying starting domains on Portraits. (c) compares the effect of adversarial training and standard training under pseudo-labeled distributions on Rotating MNIST. We use blue and red bars to represent the clean accuracy of the standardly trained model and the adversarially trained model, respectively. The gray bar indicates the correct rate of the pseudo-labels.

When we set $\zeta = 0.05$, the gradual AST with the optimal starting domain $\tau = 1$ achieves clean accuracy of $86.04\%$ ($+4.01\%$) and adversarial accuracy of $77.25\%$ ($+37.02\%$) on Portraits.

### 3.4 Training with Labeled Intermediate Domains

In the setting of GDA, the learner is provided with unlabeled intermediate domains, and the model predicts pseudo-labels on an unlabeled data set in each iteration. The pseudo-labels contain a small proportion of incorrect labels, leading to performance degradation of the trained model. In this section, we conduct experiments to investigate the optimal performance of the model trained with ground-truth labels. In other words, the learner is provided with labeled intermediate data. Although the learner has access to the ground-truth intermediate labels, we still retain the filtering process, since we need to maintain the same data size to ensure fair comparison. We present the results of the models with varying $\tau$ in Table 19 and Table 20 (due to space limitations, we attach the tables in Appendix B). Since the results in Section 3.3 show that $\zeta = 0.05$ is the optimal filtration ratio for gradual AST methods, we set $\zeta = 0.05$ in this section. From Table 19, we can see that if the learner is provided with labeled intermediate domains, the vanilla gradual self-training method ($\tau = 22$) achieves clean accuracy of $98.44\%$ on Rotating MNIST. Recall the results in Table 9 showing that the proposed gradual AST method ($\tau = 9$) achieves clean accuracy of $97.15\%$. The performance ($97.15\%$) of the proposed gradual AST is close to the optimal performance ($98.44\%$) of the gradual self-training where the learner is provided with labeled intermediate domains.

## 4 Theoretical Analysis

In this section, we theoretically analyze the experimental phenomena identified in Section 3 and explain the efficacy of gradual AST. In Section 4.1, we provide the adversarial error bound for gradual AST. In Section 4.2, we provide some theoretical insights about why AT outperforms standard training. In Section 4.3, we further present an example of a conditional Gaussian distribution to

show the error correction effect of AT on the incorrect labels. Our theory thus supports the excellent performance of gradual AST shown in Section 3.

## 4.1 Generalization Error Bounds for Gradual AST

We introduce the Margin Disparity Discrepancy (MDD) to measure the domain shifts, as below.

**Definition 4.1** (MDD [53]). Let $P$ and $Q$ be two distributions over $\mathcal{X}$. Given a scoring function class $\mathcal{F}$ and a specific scoring function $f \in \mathcal{F}$, the MDD between $P$ and $Q$ is defined as

$$d_{f,\mathcal{F}}^{(\rho)}(P,Q) \triangleq \sup_{f' \in \mathcal{F}} \left( R_Q^{(\rho)}(f',f) - R_P^{(\rho)}(f',f) \right). \tag{2}$$

When the domain shift is slight, the MDD between the domains is small. The MDD is well-defined and can be empirically estimated by the training data [53], where the estimation error can be bounded by Rademacher complexity. We next present the definition of Rademacher complexity, which is widely used in generalization theory to measure the complexity of a hypothesis class.

**Definition 4.2** (Rademacher Complexity [44]). Let $\mathcal{G}$ be a set of real-valued functions defined over $\mathcal{X}$. For a fixed collection $S$ of $n$ data points, the empirical Rademacher complexity of $\mathcal{G}$ is defined as

$$\widehat{\mathcal{R}}_S(\mathcal{G}) \triangleq \frac{2}{n} \mathbb{E}_{\varsigma} \left[ \sup_{g \in \mathcal{G}} \sum_{x_i \in S} \varsigma_i g(x_i) \right].$$

The expectation is taken over $\varsigma = (\varsigma_1, \ldots, \varsigma_n)$, where $\varsigma_i, i \in \{1, \ldots, n\}$, are independent uniform random variables taking values in $\{-1, +1\}$.

We next introduce an adversarial margin hypothesis class $\widetilde{\rho_{\mathcal{F}} \mathcal{F}}$.

**Definition 4.3** (Adversarial Margin Hypothesis Class). Given a scoring function class $\mathcal{F}$, the adversarial margin hypothesis class is defined as

$$\widetilde{\rho_{\mathcal{F}} \mathcal{F}} \triangleq \{x \mapsto \max_{\|\delta\|_q \leq \epsilon} \rho_{f'}(x + \delta, h_f(x)) : f, f' \in \mathcal{F}\}.$$

Having defined the Rademacher complexity and the MDD, we now focus on a pair of arbitrary consecutive domains. The following result shows that AST returns a new model $f'$ with bounded adversarial margin loss if the domain shifts gradually. The proof can be found in Appendix A.2.

**Theorem 4.4** (Adversarial Error Bound for AST). *Let $\langle P, f_P \rangle$ and $\langle Q, f_Q \rangle$ be two domains with gradual shifts. Suppose we have a scoring function $f$ pre-trained on domain $\langle P, f_P \rangle$, and a data set $S$ of $n$ unlabeled data points drawn i.i.d. according to distribution $Q$. $f'$ is the adapted scoring function by generated by the AST algorithm over $S$, i.e., $f' = AST(f, S)$. Then, for any $\alpha \geq 0$, the following holds with probability of at least $1 - \alpha$ over data points $S$:*

$$\widetilde{R}_Q^{(\rho)}(f') \leq \widetilde{R}_P^{(\rho)}(f) + \widetilde{\gamma}^* + \lambda^* + d_{f,\mathcal{F}}(P,Q) + \frac{2}{\rho} \widehat{\mathcal{R}}_S(\widetilde{\rho_{\mathcal{F}} \mathcal{F}}) + 6\sqrt{\frac{\log \frac{4}{\alpha}}{2n}},$$

*where $\widetilde{\gamma}^* = \min_{f' \in \mathcal{F}} \widetilde{R}_Q^{(\rho)}(f',f)$ and $\lambda^* = \min_{f \in \mathcal{F}} \left\{ R_P^{(\rho)}(f) + R_Q^{(\rho)}(f) \right\}$.*

*Remark* 4.5. Theorem 4.4 indicates that if we have a model $f$ with low adversarial margin loss on the domain $\langle P, f_P \rangle$ and the distribution shifts slightly, AST generates a new model $f'$ that is also adversarially robust on the domain $\langle Q, f_Q \rangle$. The $\lambda^*$ term is widely used in domain adaptation theory [5, 34] to implicitly characterize the conditional distribution shift. In the setting of GDA, we assume that the domain shifts only slightly, i.e., $\lambda^*$ and $d_{f,\mathcal{F}}(P,Q)$ are relatively low.

Based on the error bounds for AST, we can apply this argument inductively and sum these error differences along the domain sequence to obtain the following corollary. See proofs in Appendix A.3.

**Corollary 4.6** (Error Bounds for Gradual AST). *Given a sequence of domains $\langle P_t, f_{P_t} \rangle, t \in \{0, \ldots, T\}$ with gradual shifts, each intermediate domain has an unlabeled data set $S_t$ drawn i.i.d. from $P_t$. The model is successively trained by the AST method, i.e., $f_t = AST(f_{t-1}, S_t), t \in$*

$\{1, \dots, T\}$. *Then, for any $\alpha \geq 0$, the following holds with probability of at least $1 - \alpha$ over data points $\{S_t\}_{t=1}^T$:*

$$\widetilde{R}_{P_T}(h_{f_T}) \leq \widetilde{R}_{P_0}^{(\rho)}(f_0) + \sum_{t=1}^T \kappa_t + \frac{2T}{\rho}\widehat{\mathcal{R}}_S(\widetilde{\rho_{\mathcal{F}}\mathcal{F}}) + 6T\sqrt{\frac{\log\frac{4T}{\alpha}}{2n}},$$

*where $\kappa_t = d_{f_{t-1}, \mathcal{F}}(P_{t-1}, P_t) + \min_{f \in \mathcal{F}} \widetilde{R}_{P_t}^{(\rho)}(f, f_{t-1}) + \min_{f \in \mathcal{F}} \left\{ R_P^{(\rho)}(f) + R_Q^{(\rho)}(f) \right\}.$*

*Remark* 4.7. Our bound indicates that the adversarial risk on target domain can be controlled by the adversarial margin loss on the source domain and the discrepancy [53] between the intermediate domains. We also consider the standard risk and find that the standard risk of AST can be bounded similarly. We provide the bounds in Appendix A.4.

**Comparison with Previous Work.** Both of the works of [26] and [45] focus on binary classification and standard self-training, while our results non-trivially extend the error bounds into the setting of adversarial GDA and multiclass classification. Furthermore, [26] sets linear hypotheses as their training models, while we consider a more general hypothesis class that can be implemented by neural networks. [45] makes an assumption that the new model can always fit the pseudo-labels without error. As we can observe from the experiments, the training error tends to be small but never equal to 0. Compared to [45], we don't need the assumption, so our setting is more consistent with real experiments.

## 4.2 Adversarial Training Improves Clean Accuracy for Self-Training

AT is generally considered harmful to the clean accuracy of the model [52, 38]. However, the experimental results in Section 3 indicate that AT unexpectedly improves both the clean accuracy and adversarial accuracy of the model trained on the pseudo-labels. In this section, we provide some theoretical insights showing that when the training data contains a small portion of incorrect labels, the adversarially trained model may have a tighter generalization guarantee than the standardly trained model. Our theorem is based on the following optimization problem.

**Definition 4.8.** Given two fixed vectors $p$ and $p'$ with all positive coordinates and a fixed $N$-dimensional 0-1 vector $e$, i.e., $e \in \{0, 1\}^N$. $\Pi$ is a fixed subset of $\{1, \dots, N\}$. The variant of the Subset Sum Problem [3] can be defined as follows:

$$\min_{\widetilde{e} \in \{0,1\}^N} \left| p^T \widetilde{e} - p'^T e \right|, \quad s.t. \ \widetilde{e}_i = e_i, \forall i \in \{1, \dots, N\} \backslash \Pi.$$

We use $\Psi^*(p', p, e, \Pi)$ to denote the optimal value of the problem. Given two fixed vectors $p'$ and $e$, the value of the term $p'^T e$ is therefore constant. The optimization problem can be viewed as one of selecting a subset of $p$ such that the sum of its elements is as close as possible to $p'^T e$. Intuitively, the set $\Pi$ adjusts the difficulty of this optimization problem. For two sets $\Pi_1 \subseteq \Pi_2$, we have $\Psi^*(p', p, e, \Pi_1) \geq \Psi^*(p', p, e, \Pi_2)$ since we can optimize $\widetilde{e}$ on more coordinates.

With a little abuse of notations, in this subsection, we consider a discrete real distribution $P$ and a discrete noisy pseudo-labeled distribution $P_\eta$ over $\mathcal{X} \times \mathcal{Y}$. We denote the corresponding probability mass functions as $p$ and $p_\eta$ respectively. $p$ and $p_\eta$ can be viewed as two vectors, i.e., $p_i = p(x_i, y_i), (x_i, y_i) \in \mathcal{X} \times \mathcal{Y}, i \in \{1, \dots, |\mathcal{X} \times \mathcal{Y}|\}$. The standard risk on the real data distribution is defined as $R_P(h) = \sum_{(x,y) \in \mathcal{X} \times \mathcal{Y}} p(x, y)\mathbb{1}[h(x) \neq y]$ and the adversarial risk on the pseudo-labeled data distribution is defined as $\widetilde{R}_{P_\eta}(h) = \sum_{(x,y) \in \mathcal{X} \times \mathcal{Y}} p_\eta(x, y) \max_{\|\delta\|_q \leq \epsilon} \mathbb{1}[h(x+\delta) \neq y]$.

The next result uses the optimal value $\Psi^*(p, p_\eta, e, \Pi)$ to bound the difference between these two risks. The proof can be found in Appendix A.5.

**Theorem 4.9.** *Given two distributions $P$ and $P_\eta$ over $\mathcal{X} \times \mathcal{Y}$ with the corresponding probability mass vectors $p$ and $p_\eta$, the following bound holds for any classifier $h \in \mathcal{H}_{\mathcal{F}}$:*

$$R_P(h) \leq \widetilde{R}_{P_\eta}(h) + \Psi^*(p, p_\eta, e, \Pi_\epsilon),$$

*where $e$ is the risk vector $e_i = \mathbb{1}[h(x_i) \neq y_i]$, and $\Pi_\epsilon = \{i : \exists \|\delta_i\|_q \leq \epsilon, h(x_i + \delta_i) \neq h(x_i)\}$.*

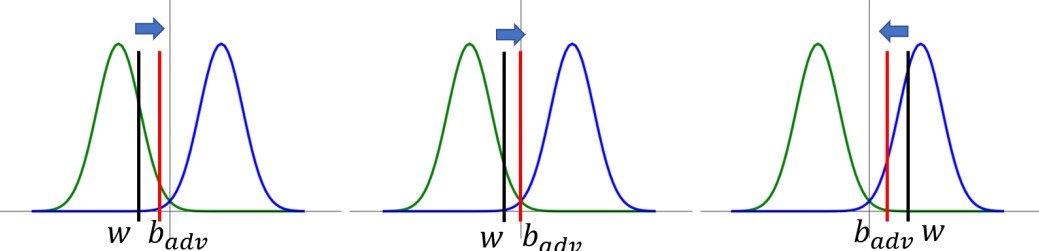

Figure 2: The visualization of the adversarially trained classifier for a conditional Gaussian distribution. AT consistently moves the threshold $b_{adv}$ close to the optimal threshold $b = 0$, leading to better generalization on the real data distribution. Best viewed in color.

By setting $\epsilon = 0$, we can derive the error bound for standard training: $R_P(h) \leq R_{P_\eta}(h) + \Psi^*(p, p_\eta, e, \Pi_0)$. By the definition of the set $\Pi_\epsilon$, the larger the perturbation radius $\epsilon$, the larger the size of $\Pi_\epsilon$, i.e., $\Pi_0 \subseteq \Pi_\epsilon$. We then have $\Psi^*(p, p_\eta, e, \Pi_\epsilon) \leq \Psi^*(p, p_\eta, e, \Pi_0)$. Theorem 4.9 implies that if the adversarial risk $\widetilde{R}_{P_\eta}(h)$ on the pseudo-labeled distribution is well controlled by AT, then AT may yield a tighter generalization guarantee than standard training.

### 4.3 A Toy Example of a Conditional Gaussian Distribution

We further provide an example of a conditional Gaussian distribution to show the efficacy of AT under incorrect pseudo-labels.

**Example 4.10** (Conditional Gaussian Distribution). We consider the input space $\mathcal{X} = \mathbb{R}$ and the label space $\mathcal{Y} = \{-1, +1\}$. For simplicity, we assume that the two classes have symmetrical mean parameters $\{-\mu, +\mu\}$ and the same standard deviations $\sigma$. The $(\mu, \sigma)$-conditional Gaussian distribution $P$ can then be defined as a distribution over $\mathcal{X} \times \mathcal{Y} = \mathbb{R} \times \{-1, +1\}$, where the conditional distribution of $X$ given $Y$ is $X|Y = y \sim \mathcal{N}(y\mu, \sigma^2)$ and the marginal distribution of $Y$ is $P_Y(\{-1\}) = P_Y(\{+1\}) = 1/2$. We consider a threshold hypothesis class, i.e., $\mathcal{H} \triangleq \{x \mapsto \text{sgn}(x - b) : b \in \mathbb{R}\}$, where sgn is the sign function. We denote the hypothesis with threshold $b$ as $h_b$. In the setting of self-training, we have a labeling function $h_w$, which is the model trained on the previous domain. The threshold $w \neq 0$, since there is a distribution shift. The function $h_w$ assigns $y = -1$ to the data points where $x < w$ and assigns $y = +1$ to the data points where $x > w$. We denote the pseudo-labeled data distribution over $\mathcal{X} \times \mathcal{Y}$ as $P_\eta$. We use $R_{P_\eta}(h)$ and $\widetilde{R}_{P_\eta}(h)$ to respectively denote the standard 0-1 risk and adversarial 0-1 risk on the pseudo-labeled data distribution $P_\eta$. We assume $|w| \ll \mu$ and $\epsilon \ll \mu$, since the domain shift is slight and the adversarial perturbation is subtle. We also assume $\mu > \sigma$, which means that most samples can be well separated. The learner is provided with infinite samples. Under these assumptions, we provide the following results to demonstrate the efficacy of AT. The proof can be found in Appendix A.6.

**Theorem 4.11.** *For the real data distribution $P$ defined in Example 4.10, the optimal classifier that minimizes the standard risk $R_P(h)$ and adversarial risk $\widetilde{R}_P(h)$ is $h_0(\cdot) = \text{sgn}(\cdot)$. For the pseudo-labeled distribution $P_\eta$ defined in Example 4.10, the standardly trained classifier $h_{std}$ that minimizes the standard risk $R_{P_\eta}(h)$ is $h_{std} = h_w$; the adversarially trained classifier $h_{adv}$ that minimizes the adversarial risk $\widetilde{R}_{P_\eta}(h)$ has the following corresponding threshold $b_{adv}$,*

$$b_{adv} = \begin{cases} w + \epsilon, & \text{if } w < -\epsilon \\ 0, & \text{if } -\epsilon < w < \epsilon \\ w - \epsilon, & \text{if } w > \epsilon \end{cases}$$

*Remark* 4.12. Theorem 4.11 shows that the adversarially trained model $h_{adv}$ consistently moves the threshold close to the optimal threshold $b = 0$, leading to a smaller generalization error. We visually represent the adversarially trained model in Figure 2.

## 5 Related Work

**Self-Training.** A series of works have achieved significant progress using self-training in semi-supervised learning [29, 40] and domain adaptation [56]. The recent work [9] has proposed robust

self-training (RST) to leverage additional unlabeled data for robustness [2, 36]. While RST [9] focuses on utilizing additional unlabeled data drawn from the same distribution to improve adversarial robustness, our work investigates the different effects of vanilla self-training and adversarial self-training on clean accuracy.

**Gradual Domain Adaptation.** Unlike the UDA problem, GDA has intermediate domains between the source domain and target domain of UDA. Some works [22, 17, 48] propose various algorithms to handle these evolving domains in a computer vision context. The recent work [26] adopts the self-training method and proposes gradual self-training, which iteratively adapts the model along the domain sequence. Moreover, [26] provides the first learning theory for GDA and investigates when and why the gradual structure helps. [45] further improves the generalization error bound in [26] and reveals the influence of the number of intermediate domains on the error bound. [10] investigates a more difficult setting in which the learner is provided with a set of unordered intermediate data. The authors propose TDOL, a framework that generates synthetic domains by indexing the intermediate data. However, the aforementioned works pay insufficient attention to the adversarial robustness of the model in a GDA context. In the field of UDA, the work of [11] constructs synthetic intermediate domains by creating Grassmannian manifolds between the source and the target. Another line of work instead uses a generative network, such as a cycle generative adversarial network [23], to generate the intermediate data.

**Adversarial Training.** After [41] shows that DNNs are fragile to adversarial attacks, a large amount of works have proposed various attack methods [33, 27, 8] and defense methods [47, 19, 51, 31, 20, 24, 30]. Adversarial training [19] is one of the popular and effective methods that improves adversarial robustness by adding adversarial examples to the training dataset. From the theoretical perspective, some works focus on the sample complexity [50, 54, 12] and the generalization of adversarial training. [32] investigates the trade-off between robustness and fairness. [46, 55] study adversarial robustness under self-supervised learning.

# 6 Conclusion

In this work, we empirically demonstrate that gradual AST improves both the clean accuracy and the adversarial accuracy of the GDA model. We reveal that the reason for this performance improvement is that AT outperforms standard training when the training data contains incorrect pseudo-labels. We first provide generalization error bounds for gradual AST in a multiclass setting. We then construct the Subset Sum Problem to connect the adversarial error and the standard error, providing theoretical insights into the superiority of AST in GDA. The example of conditional Gaussian distribution is further provided to give additional insights into the efficacy of AT on pseudo-labeled training data.

# Acknowledgements

This work is supported by the National Natural Science Foundation of China under Grant 61976161, the Fundamental Research Funds for the Central Universities under Grant 2042022rc0016.

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
