# Appendix

## A    Theory

In this section, we show the proofs of the results in the main body.

### A.1    Triangle Inequality of the Expected Adversarial Margin Loss

**Lemma A.1.** *For any distribution $P$ over $\mathcal{X}$, the expected adversarial margin loss $\widetilde{R}_P^{(\rho)}$ defined in Eq. (1) satisfies the triangle inequality, i.e., for any scoring functions $f_1, f_2, f_3 \in \mathcal{F}$, the following inequalities hold:*

$$\widetilde{R}_P^{(\rho)}(f_1, f_3) \leq \widetilde{R}_P^{(\rho)}(f_1, f_2) + \widetilde{R}_P^{(\rho)}(f_2, f_3)$$

$$\widetilde{R}_P^{(\rho)}(f_1, f_2) \leq \widetilde{R}_P^{(\rho)}(f_1, f_3) + \widetilde{R}_P^{(\rho)}(f_2, f_3)$$

*Proof of Lemma A.1.* Recall that $\widetilde{R}_P^{(\rho)}(f, f') \triangleq \underset{x \sim P}{\mathbb{E}} \max_{\|\delta\|_q \leq \epsilon} \Theta_\rho \circ \rho_f(x + \delta, h_{f'}(x))$.

(I). For the first inequality, we prove that for any data point $x \in \mathcal{X}$, the following holds:

$$\max_{\|\delta\|_q \leq \epsilon} \Theta_\rho \circ \rho_{f_1}(x+\delta, h_{f_3}(x)) \leq \max_{\|\delta\|_q \leq \epsilon} \Theta_\rho \circ \rho_{f_1}(x+\delta, h_{f_2}(x)) + \max_{\|\delta\|_q \leq \epsilon} \Theta_\rho \circ \rho_{f_2}(x+\delta, h_{f_3}(x)) \quad (3)$$

We prove it from two possible cases:

- If there exits a perturbation $\|\delta\|_q \leq \epsilon$, such that $h_{f_1}(x+\delta) \neq h_{f_2}(x)$ or $h_{f_2}(x+\delta) \neq h_{f_3}(x)$, then $\Theta_\rho \circ \rho_{f_1}(x+\delta, h_{f_2}(x)) + \Theta_\rho \circ \rho_{f_2}(x+\delta, h_{f_3}(x)) \geq 1$ by the definition of the expected adversarial margin loss. On the other hand, $\Theta_\rho \circ \rho_{f_1}(x + \delta, h_{f_3}(x)) \leq 1$, since $\Theta_\rho(m) \leq 1$ for any $m$. Thus the inequality holds.

- If for any perturbation $\|\delta\|_q \leq \epsilon$, we have $h_{f_1}(x+\delta) = h_{f_2}(x)$ and $h_{f_2}(x+\delta) = h_{f_3}(x)$, by setting $\delta = 0$, we then obtain that $h_{f_2}(x) = h_{f_3}(x)$. Then we can derive that $h_{f_1}(x + \delta) = h_{f_2}(x) = h_{f_3}(x)$ for any $\|\delta\|_q \leq \epsilon$, and

$$\max_{\|\delta\|_q \leq \epsilon} \Theta_\rho \circ \rho_{f_1}(x + \delta, h_{f_3}(x))$$
$$\leq \max_{\|\delta\|_q \leq \epsilon} \Theta_\rho \circ \rho_{f_1}(x + \delta, h_{f_3}(x)) + \max_{\|\delta\|_q \leq \epsilon} \Theta_\rho \circ \rho_{f_2}(x + \delta, h_{f_3}(x))$$
$$\leq \max_{\|\delta\|_q \leq \epsilon} \Theta_\rho \circ \rho_{f_1}(x + \delta, h_{f_2}(x)) + \max_{\|\delta\|_q \leq \epsilon} \Theta_\rho \circ \rho_{f_2}(x + \delta, h_{f_3}(x))$$

Then take expectation over the distribution $P$ on both sides of Eq. (3) and we get the first inequality.

(II). For the second inequality, we prove it similarly. We first prove that for any data point $x \in \mathcal{X}$, the following holds:

$$\max_{\|\delta\|_q \leq \epsilon} \Theta_\rho \circ \rho_{f_1}(x+\delta, h_{f_2}(x)) \leq \max_{\|\delta\|_q \leq \epsilon} \Theta_\rho \circ \rho_{f_1}(x+\delta, h_{f_3}(x)) + \max_{\|\delta\|_q \leq \epsilon} \Theta_\rho \circ \rho_{f_2}(x+\delta, h_{f_3}(x)) \quad (4)$$

We prove it from two possible cases:

- If there exits a perturbation $\|\delta\|_q \leq \epsilon$, such that $h_{f_2}(x + \delta) \neq h_{f_3}(x)$, then $\Theta_\rho \circ \rho_{f_2}(x + \delta, h_{f_3}(x)) = 1$ by the definition of the expected adversarial margin loss. On the other hand, $\Theta_\rho \circ \rho_{f_1}(x + \delta, h_{f_2}(x)) \leq 1$, since $\Theta_\rho(m) \leq 1$ for any $m$. Thus the inequality holds.

- If for any perturbation $\|\delta\|_q \leq \epsilon$, we have $h_{f_2}(x + \delta) = h_{f_3}(x)$, by setting $\delta = 0$, we then obtain that $h_{f_2}(x) = h_{f_3}(x)$. We can derive that

$$\max_{\|\delta\|_q \leq \epsilon} \Theta_\rho \circ \rho_{f_1}(x + \delta, h_{f_2}(x))$$
$$\leq \max_{\|\delta\|_q \leq \epsilon} \Theta_\rho \circ \rho_{f_1}(x + \delta, h_{f_2}(x)) + \max_{\|\delta\|_q \leq \epsilon} \Theta_\rho \circ \rho_{f_2}(x + \delta, h_{f_3}(x))$$
$$\leq \max_{\|\delta\|_q \leq \epsilon} \Theta_\rho \circ \rho_{f_1}(x + \delta, h_{f_3}(x)) + \max_{\|\delta\|_q \leq \epsilon} \Theta_\rho \circ \rho_{f_2}(x + \delta, h_{f_3}(x))$$

Then take expectation over the distribution $P$ on both sides of Eq. (3) and we get the second inequality. $\qquad\square$

## A.2 Proofs of Theorem 4.4

**Theorem 4.4.** *Let $\langle P, f_P \rangle$ and $\langle Q, f_Q \rangle$ be two domains with gradual shifts. Suppose we have a scoring function $f$ pre-trained on domain $\langle P, f_P \rangle$, and a data set $S$ of $n$ unlabeled data points drawn i.i.d. according to distribution $Q$. $f'$ is the adapted scoring function by AST algorithm over $S$, i.e., $f' = AST(f, S)$. Then, for any $\alpha \geq 0$, the following holds with probability of at least $1 - \alpha$ over data points $S$:*

$$\widetilde{R}_Q^{(\rho)}(f') \leq \widetilde{R}_P^{(\rho)}(f) + \widetilde{\gamma}^* + \lambda^* + d_{f,\mathcal{F}}(P,Q) + \frac{2}{\rho}\widehat{\mathcal{R}}_S(\widetilde{\rho_{\mathcal{F}}\mathcal{F}}) + 6\sqrt{\frac{\log\frac{4}{\alpha}}{2n}},$$

*where $\widetilde{\gamma}^* = \min\limits_{f'\in\mathcal{F}}\widetilde{R}_Q^{(\rho)}(f', f)$ and $\lambda^* = \min\limits_{f\in\mathcal{F}}R_P^{(\rho)}(f) + R_Q^{(\rho)}(f)$.*

Before we present the proof of the theorem, we first provide some lemmas.

We now provide a lemma which uses the Rademacher complexity to connect the population and empirical error.

**Lemma A.2** (Rademacher Bound [4]). *Suppose that $\mathcal{G}$ is a class of functions mapping $\mathcal{X}$ to [0,1]. Then, for any $\alpha > 0$, with probability at least $1 - \alpha$ over samples $S$ of size $n$, the following holds for all $g \in \mathcal{G}$:*

$$\mathbb{E}_{x\sim P}\, g(x) \leq \mathbb{E}_{x\sim\widehat{P}}\, g(x) + \widehat{\mathcal{R}}_S(\mathcal{G}) + 3\sqrt{\frac{\log\frac{2}{\alpha}}{2n}}, \tag{5}$$

*where $\mathbb{E}_{x\sim P}\, g(x)$ is the expectation of a function $g$, and $\mathbb{E}_{x\sim\widehat{P}}\, g(x)$ is its empirical average over the samples $S$ drawn i.i.d. according to the distribution $P$.*

The next lemma says that the empirically trained classifier gets closer to the optimal classifier that minimizes the population error when the sample size is large. We characterize this by bounding the generalization error of the classifier.

**Lemma A.3.** *Let $f' \in \mathcal{F}$ be a scoring function and $P$ be a distribution over $\mathcal{X}$. Let $\hat{f}$ be the empirically trained model: $\hat{f} = \arg\min\limits_{f\in\mathcal{F}}\widetilde{R}_{\widehat{P}}^{(\rho)}$, and $f^*$ be the optimal model that minimizes the population error: $f^* = \arg\min\limits_{f\in\mathcal{F}}\widetilde{R}_P^{(\rho)}$. Then, for any $\alpha \geq 0$, the following holds with the probability at least $1 - \alpha$ over the data set $S$ of size $n$,*

$$\widetilde{R}_P^{(\rho)}(\hat{f}, f') \leq \widetilde{R}_P^{(\rho)}(f^*, f') + \frac{2}{\rho}\widehat{\mathcal{R}}_S(\widetilde{\rho_{\mathcal{F}}\mathcal{F}}) + 6\sqrt{\frac{\log\frac{4}{\alpha}}{2n}},$$

*where $\widetilde{\rho_{\mathcal{F}}\mathcal{F}} \triangleq \{x \mapsto \max\limits_{\|\delta\|_q\leq\epsilon}\rho_{f'}(x+\delta, h_f(x)) : f, f' \in \mathcal{F}\}$.*

*Proof of Lemma A.3.* By applying Lemma A.2, the following holds with probability at least $1 - \alpha$:

$$
\begin{aligned}
&\widetilde{R}_P^{(\rho)}(\hat{f}, f')\\
\leq&\widetilde{R}_P^{(\rho)}(\hat{f}, f') - \widetilde{R}_{\widehat{P}}^{(\rho)}(\hat{f}, f') + \widetilde{R}_{\widehat{P}}^{(\rho)}(\hat{f}, f')\\
\leq&\widehat{\mathcal{R}}_S(\Theta_\rho\circ\widetilde{\rho_{\mathcal{F}}\mathcal{F}}) + 3\sqrt{\frac{\log\frac{2}{\alpha}}{2n}} + \widetilde{R}_{\widehat{P}}^{(\rho)}(\hat{f}, f')
\end{aligned}
\tag{6}
$$

Apply Lemma A.2 again, the following holds with probability at least $1 - \alpha$:

$$\widehat{\mathcal{R}}_S(\Theta_\rho \circ \widetilde{\rho_\mathcal{F}\mathcal{F}}) + 3\sqrt{\frac{\log \frac{2}{\alpha}}{2n}} + \widetilde{R}_{\widehat{P}}^{(\rho)}(\hat{f}, f')$$

$$\leq \widehat{\mathcal{R}}_S(\Theta_\rho \circ \widetilde{\rho_\mathcal{F}\mathcal{F}}) + 3\sqrt{\frac{\log \frac{2}{\alpha}}{2n}} + \widetilde{R}_{\widehat{P}}^{(\rho)}(\hat{f}, f') - \widetilde{R}_{\widehat{P}}^{(\rho)}(f^*, f') + \widetilde{R}_{\widehat{P}}^{(\rho)}(f^*, f')$$

$$\overset{(i)}{\leq} \widehat{\mathcal{R}}_S(\Theta_\rho \circ \widetilde{\rho_\mathcal{F}\mathcal{F}}) + 3\sqrt{\frac{\log \frac{2}{\alpha}}{2n}} + \widetilde{R}_{\widehat{P}}^{(\rho)}(f^*, f') \tag{7}$$

$$\leq \widehat{\mathcal{R}}_S(\Theta_\rho \circ \widetilde{\rho_\mathcal{F}\mathcal{F}}) + 3\sqrt{\frac{\log \frac{2}{\alpha}}{2n}} + \widetilde{R}_{\widehat{P}}^{(\rho)}(f^*, f') - \widetilde{R}_{P}^{(\rho)}(f^*, f') + \widetilde{R}_{P}^{(\rho)}(f^*, f')$$

$$\overset{(ii)}{\leq} 2\widehat{\mathcal{R}}_S(\Theta_\rho \circ \widetilde{\rho_\mathcal{F}\mathcal{F}}) + 6\sqrt{\frac{\log \frac{2}{\alpha}}{2n}} + \widetilde{R}_{P}^{(\rho)}(f^*, f')$$

where $(i)$ is because $\hat{f} = \arg\min_{f \in \mathcal{F}} \widetilde{R}_{\widehat{P}}^{(\rho)}$ and $(ii)$ is from Lemma A.2.

Combining Eq. (6) and Eq. (7) using union bound, the following holds with probability at least $1 - 2\alpha'$:

$$\widetilde{R}_{P}^{(\rho)}(\hat{f}, f') \leq 2\widehat{\mathcal{R}}_S(\Theta_\rho \circ \widetilde{\rho_\mathcal{F}\mathcal{F}}) + 6\sqrt{\frac{\log \frac{2}{\alpha'}}{2n}} + \widetilde{R}_{P}^{(\rho)}(f^*, f')$$

Set $\alpha' = \alpha/2$ and the following holds with probability at least $1 - \alpha$:

$$\widetilde{R}_{P}^{(\rho)}(\hat{f}, f') \leq 2\widehat{\mathcal{R}}_S(\Theta_\rho \circ \widetilde{\rho_\mathcal{F}\mathcal{F}}) + 6\sqrt{\frac{\log \frac{4}{\alpha}}{2n}} + \widetilde{R}_{P}^{(\rho)}(f^*, f').$$

Since the function $\Theta_\rho$ is $\frac{1}{\rho}$-Lipschitz, by Talagrand's contraction Lemma [28], $\widehat{\mathcal{R}}_S(\Theta_\rho \circ \widetilde{\rho_\mathcal{F}\mathcal{F}})$ is bounded by $\frac{1}{\rho}\widehat{\mathcal{R}}_S(\widetilde{\rho_\mathcal{F}\mathcal{F}})$. Thus we have:

$$\widetilde{R}_{P}^{(\rho)}(\hat{f}, f') \leq \widetilde{R}_{P}^{(\rho)}(f^*, f') + \frac{2}{\rho}\widehat{\mathcal{R}}_S(\widetilde{\rho_\mathcal{F}\mathcal{F}}) + 6\sqrt{\frac{\log \frac{4}{\alpha}}{2n}}.$$

$$\square$$

The next lemma shows that the MDD can control the error difference of the model on different domains.

**Lemma A.4.** *Let $\langle P, f_P \rangle$ and $\langle Q, f_Q \rangle$ be two domains. For any scoring function $f \in \mathcal{F}$, the following holds:*

$$R_Q(h_f) \leq R_{P}^{(\rho)}(f) + \lambda^* + d_{f,\mathcal{F}}^{(\rho)}(P, Q) \leq \widetilde{R}_{P}^{(\rho)}(f) + \lambda^* + d_{f,\mathcal{F}}^{(\rho)}(P, Q),$$

*where $\lambda^* = \min_{f \in \mathcal{F}}\{R_{P}^{(\rho)}(f) + R_{Q}^{(\rho)}(f)\}$.*

*Proof of Lemma A.4.* Recall that $R_Q(h_f) = R_Q(h_f, h_Q)$ and $R_{P}^{(\rho)}(f) = R_{P}^{(\rho)}(f, f_P)$. Let $f^*$ be the optimal that achieves minimal margin loss on both $P$ and $Q$: $f^* = \arg\min_{f \in \mathcal{F}}\{R_{P}^{(\rho)}(f) + R_{Q}^{(\rho)}(f)\}$, then we have:

$$
\begin{aligned}
R_Q(h_f) &= R_Q(h_f, h_Q) \\
&= \mathop{\mathbb{E}}_{x \sim Q} \mathbb{1}[h_f(x) \neq h_Q(x)] \\
&\leq \mathop{\mathbb{E}}_{x \sim Q} \mathbb{1}[h_{f^*}(x) \neq h_f(x)] + \mathop{\mathbb{E}}_{x \sim Q} \mathbb{1}[h_{f^*}(x) \neq h_Q(x)] \\
&\leq R_{Q}^{(\rho)}(f^*, f) + R_{Q}^{(\rho)}(f^*, f_Q) \\
&\leq R_{Q}^{(\rho)}(f^*, f) + R_{Q}^{(\rho)}(f^*, f_Q) + R_{P}^{(\rho)}(f^*, f) - R_{P}^{(\rho)}(f^*, f) \\
&\leq R_{Q}^{(\rho)}(f^*, f_Q) + R_{P}^{(\rho)}(f^*, f) + d_{f,\mathcal{F}}^{(\rho)}(P, Q)
\end{aligned}
\tag{8}
$$

where the last inequality is from the definition of the MDD in Eq. (2). By setting $\epsilon = 0$ in Lemma A.1, we can get that the margin loss satisfies the triangle inequality. Then we have:

$$
\begin{aligned}
& R_Q^{(\rho)}(f^*, f_Q) + R_P^{(\rho)}(f^*, f) + d_{f,\mathcal{F}}^{(\rho)}(P, Q) \\
\leq & R_Q^{(\rho)}(f^*, f_Q) + R_P^{(\rho)}(f^*, f_P) + R_P^{(\rho)}(f, f_P) + d_{f,\mathcal{F}}^{(\rho)}(P, Q) \\
\leq & R_P^{(\rho)}(f, f_P) + R_Q^{(\rho)}(f^*, f_Q) + R_P^{(\rho)}(f^*, f_P) + d_{f,\mathcal{F}}^{(\rho)}(P, Q) \\
= & R_P^{(\rho)}(f) + \lambda^* + d_{f,\mathcal{F}}^{(\rho)}(P, Q)
\end{aligned}
\tag{9}
$$

where $\lambda^* = \min\limits_{f \in \mathcal{F}} \{R_P^{(\rho)}(f) + R_Q^{(\rho)}(f)\}$.

Combining (8) and (9), we then get :

$$
R_Q(h_f) \leq R_P^{(\rho)}(f) + \lambda^* + d_{f,\mathcal{F}}^{(\rho)}(P, Q).
$$

Finally, since the adversarial margin loss is larger than the standard margin loss, we have:

$$
R_Q(h_f) \leq \widetilde{R}_P^{(\rho)}(f) + \lambda^* + d_{f,\mathcal{F}}^{(\rho)}(P, Q).
$$

$\square$

The next lemma says that if we have two labeling functions with similar prediction, then for any scoring function $f$, the adversarial margin loss of $f$ according to these two labeling functions are close.

**Lemma A.5.** *Let $f, f_1, f_2 \in \mathcal{F}$ be scoring functions and $P$ be a distribution over $\mathcal{X}$. Then we have*

$$
\widetilde{R}_P^{(\rho)}(f, f_1) \leq \widetilde{R}_P^{(\rho)}(f, f_2) + R_P(f_1, f_2)
$$

*Proof of Lemma A.5.* By the definition of the expected adversarial margin loss,

$$
\begin{aligned}
\widetilde{R}_P^{(\rho)}(f, f_1) &= \mathop{\mathbb{E}}_{x \sim P} \max_{\|\delta\|_q \leq \epsilon} \Theta_\rho \circ \rho_f(x + \delta, h_{f_1}(x)) \\
&= \mathop{\mathbb{E}}_{x \sim P} \max_{\|\delta\|_q \leq \epsilon} \Theta_\rho \circ \rho_f(x + \delta, h_{f_1}(x)) \cdot \mathbb{1}[h_{f_1}(x) = h_{f_2}(x)] \\
&\quad + \mathop{\mathbb{E}}_{x \sim P} \max_{\|\delta\|_q \leq \epsilon} \Theta_\rho \circ \rho_f(x + \delta, h_{f_1}(x)) \cdot \mathbb{1}[h_{f_1}(x) \neq h_{f_2}(x)] \\
&\leq \mathop{\mathbb{E}}_{x \sim P} \max_{\|\delta\|_q \leq \epsilon} \Theta_\rho \circ \rho_f(x + \delta, h_{f_1}(x)) \cdot \mathbb{1}[h_{f_1}(x) = h_{f_2}(x)] + \mathop{\mathbb{E}}_{x \sim P} \mathbb{1}[h_{f_1}(x) \neq h_{f_2}(x)] \\
&\leq \mathop{\mathbb{E}}_{x \sim P} \max_{\|\delta\|_q \leq \epsilon} \Theta_\rho \circ \rho_f(x + \delta, h_{f_2}(x)) \cdot \mathbb{1}[h_{f_1}(x) = h_{f_2}(x)] + \mathop{\mathbb{E}}_{x \sim P} \mathbb{1}[h_{f_1}(x) \neq h_{f_2}(x)] \\
&\leq \mathop{\mathbb{E}}_{x \sim P} \max_{\|\delta\|_q \leq \epsilon} \Theta_\rho \circ \rho_f(x + \delta, h_{f_2}(x)) + \mathop{\mathbb{E}}_{x \sim P} \mathbb{1}[h_{f_1}(x) \neq h_{f_2}(x)] \\
&= \widetilde{R}_P^{(\rho)}(f, f_2) + R_P(f_1, f_2)
\end{aligned}
$$

$\square$

Based on the previous lemmas, we now present our proof of Theorem 4.4.

***Proof of Theorem 4.4.*** We denote the optimal scoring function that minimizes the adversarial margin loss on the pseudo-labeled distribution of $Q$ as $f^* = \arg\min\limits_{f' \in \mathcal{F}} \widetilde{R}_Q^{(\rho)}(f', f)$. The adversarial margin loss of $f^*$ is $\widetilde{\gamma}^* = \widetilde{R}_Q^{(\rho)}(f^*, f) = \min\limits_{f' \in \mathcal{F}} \widetilde{R}_Q^{(\rho)}(f', f)$.

By Lemma A.4, we have

$$
R_Q(h_f) \leq \widetilde{R}_P^{(\rho)}(f) + \lambda^* + d_{f,\mathcal{F}}^{(\rho)}(P, Q),
\tag{10}
$$

where $R_Q(h_f)$ represents the labeling error of the function $h_f$ on domain $\langle Q, h_Q \rangle$.

Then, by Lemma A.3, the empirically trained model $f'$ on the pseudo-labeled distribution has a controlled expected adversarial margin loss. The following holds with probability at least $1 - \alpha$:

$$
\begin{aligned}
\widetilde{R}_Q^{(\rho)}(f', f) &\leq \min_{f' \in \mathcal{F}} \widetilde{R}_Q^{(\rho)}(f', f) + \frac{2}{\rho} \widehat{\mathcal{R}}_S(\widetilde{\rho_{\mathcal{F}} \mathcal{F}}) + 6 \sqrt{\frac{\log \frac{4}{\alpha}}{2n}} \\
&= \widetilde{\gamma}^* + \frac{2}{\rho} \widehat{\mathcal{R}}_S(\widetilde{\rho_{\mathcal{F}} \mathcal{F}}) + 6 \sqrt{\frac{\log \frac{4}{\alpha}}{2n}}
\end{aligned}
\tag{11}
$$

Finally, by Lemma A.5, we have

$$
\begin{aligned}
\widetilde{R}_Q^{(\rho)}(f', f_Q) &\leq \widetilde{R}_Q^{(\rho)}(f', f) + R_Q(h_f) \\
&\leq \widetilde{R}_P^{(\rho)}(f) + \widetilde{\gamma}^* + \lambda^* + d_{f,\mathcal{F}}(P, Q) + \frac{2}{\rho} \widehat{\mathcal{R}}_S(\widetilde{\rho_{\mathcal{F}} \mathcal{F}}) + 6 \sqrt{\frac{\log \frac{4}{\alpha}}{2n}},
\end{aligned}
$$

where the last inequality is from (10) and (11). $\qquad \square$

### A.3 Proofs of Corollary 4.6

**Corollary 4.6.** *Given a sequence of domains $\langle P_t, f_{P_t} \rangle, t \in \{0, \ldots, T\}$ with gradual shifts, each intermediate domain has an unlabeled data set $S_t$ drawn i.i.d. from $P_t$. The model is successively trained by AST method, i.e., $f_t = AST(f_{t-1}, S_t), t \in \{1, \ldots, T\}$. Then, for any $\alpha \geq 0$, the following holds with probability of at least $1 - \alpha$ over the unlabeled data points $\{S_t\}_{t=1}^T$,*

$$
\widetilde{R}_{P_T}(h_{f_T}) \leq \widetilde{R}_{P_0}^{(\rho)}(f_0) + \sum_{t=1}^T \kappa_t + \frac{2T}{\rho} \widehat{\mathcal{R}}_S(\widetilde{\rho_{\mathcal{F}} \mathcal{F}}) + 6T \sqrt{\frac{\log \frac{4T}{\alpha}}{2n}},
$$

*where $\kappa_t = d_{f_{t-1}, \mathcal{F}}(P_{t-1}, P_t) + \min_{f \in \mathcal{F}} \widetilde{R}_{P_t}^{(\rho)}(f, f_{t-1}) + \min_{f \in \mathcal{F}} \left\{ R_{P_{t-1}}^{(\rho)}(f) + R_{P_t}^{(\rho)}(f) \right\}.$*

*Proof of Corollary 4.6.* By Theorem 4.4, the following holds for any $t \in \{1, \ldots, T\}$ with probability at least $1 - \alpha$:

$$
\widetilde{R}_{P_t}^{(\rho)}(f_t) \leq \widetilde{R}_{P_{t-1}}^{(\rho)}(f_{t-1}) + \widetilde{\gamma}^* + \lambda^* + d_{f_{t-1}, \mathcal{F}}(P_{t-1}, P_t) + \frac{2}{\rho} \widehat{\mathcal{R}}_S(\widetilde{\rho_{\mathcal{F}} \mathcal{F}}) + 6 \sqrt{\frac{\log \frac{4}{\alpha}}{2n}},
$$

where $\widetilde{\gamma}^* = \min_{f \in \mathcal{F}} \widetilde{R}_{P_t}^{(\rho)}(f, f_{t-1})$ and $\lambda^* = \min_{f \in \mathcal{F}} \left\{ R_{P_{t-1}}^{(\rho)}(f) + R_{P_t}^{(\rho)}(f) \right\}$.

By summing this inequality over $t \in \{1, \ldots, T\}$ and applying union bound, the following holds with probability at least $1 - \alpha$:

$$
\widetilde{R}_{P_T}^{(\rho)}(f_T) \leq \widetilde{R}_{P_0}^{(\rho)}(f_0) + \sum_{t=1}^T \kappa_t + \frac{2T}{\rho} \widehat{\mathcal{R}}_S(\widetilde{\rho_{\mathcal{F}} \mathcal{F}}) + 6T \sqrt{\frac{\log \frac{4T}{\alpha}}{2n}},
$$

Finally, the margin loss can be lower bounded by the 0-1 loss, i.e., $\widetilde{R}_{P_T}(h_{f_T}) \leq \widetilde{R}_{P_T}^{(\rho)}(f_T)$ hence we get the result:

$$
\widetilde{R}_{P_T}(h_{f_T}) \leq \widetilde{R}_{P_0}^{(\rho)}(f_0) + \sum_{t=1}^T \kappa_t + \frac{2T}{\rho} \widehat{\mathcal{R}}_S(\widetilde{\rho_{\mathcal{F}} \mathcal{F}}) + 6T \sqrt{\frac{\log \frac{4T}{\alpha}}{2n}},
$$

where $\kappa_t = d_{f_{t-1}, \mathcal{F}}(P_{t-1}, P_t) + \min_{f \in \mathcal{F}} \widetilde{R}_{P_t}^{(\rho)}(f, f_{t-1}) + \min_{f \in \mathcal{F}} \left\{ R_{P_{t-1}}^{(\rho)}(f) + R_{P_t}^{(\rho)}(f) \right\}.$ $\qquad \square$

## A.4 Standard Error Bounds for AST

**Theorem A.6.** *Let $\langle P, f_P \rangle$ and $\langle Q, f_Q \rangle$ be two domains with gradual shifts. $f, f'$, and $S$ are defined as the same as in Theorem 4.4. Then, for any $\alpha \geq 0$, the following holds with probability of at least $1 - \alpha$ over data points $S$:*

$$R_Q^{(\rho)}(f') \leq R_P^{(\rho)}(f) + \widetilde{\gamma}^* + \lambda^* + d_{f,\mathcal{F}}(P, Q) + \frac{2}{\rho}\widehat{\mathcal{R}}_S(\widetilde{\rho_{\mathcal{F}}\mathcal{F}}) + 6\sqrt{\frac{\log\frac{4}{\alpha}}{2n}},$$

*where $\widetilde{\gamma}^* = \min\limits_{f'\in\mathcal{F}} \widetilde{R}_Q^{(\rho)}(f', f)$ and $\lambda^* = \min\limits_{f\in\mathcal{F}}\left\{ R_P^{(\rho)}(f) + R_Q^{(\rho)}(f)\right\}$.*

*Proof of Theorem A.6.* Recall that $f' = AST(f, S) = \arg\min\limits_{f'\in\mathcal{F}}\widetilde{R}_Q^{(\rho)}(f', f)$. We denote the optimal scoring function that minimizes the adversarial margin loss on the pseudo-labeled distribution of $Q$ as $f^* = \arg\min\limits_{f'\in\mathcal{F}}\widetilde{R}_Q^{(\rho)}(f', f)$. The adversarial margin loss of $f^*$ is $\widetilde{\gamma}^* = \widetilde{R}_Q^{(\rho)}(f^*, f) = \min\limits_{f'\in\mathcal{F}}\widetilde{R}_Q^{(\rho)}(f', f)$. Then we have

$$
\begin{aligned}
R_Q^{(\rho)}(f') - R_P^{(\rho)}(f) &\leq R_Q^{(\rho)}(f') - R_Q(f) + R_Q(f) - R_P^{(\rho)}(f) \\
&\overset{(i)}{\leq} \widetilde{R}_Q^{(\rho)}(f') - R_Q(f) + R_Q(f) - R_P^{(\rho)}(f) \\
&\overset{(ii)}{\leq} \widetilde{R}_Q^{(\rho)}(f') - R_Q(f) + \lambda^* + d_{f,\mathcal{F}}^{(\rho)}(P, Q) \\
&\overset{(iii)}{\leq} \widetilde{R}_Q^{(\rho)}(f', f_Q) + \widetilde{R}_Q^{(\rho)}(f', f) - \widetilde{R}_Q^{(\rho)}(f', f_Q) + \lambda^* + d_{f,\mathcal{F}}^{(\rho)}(P, Q) \\
&\leq \widetilde{R}_Q^{(\rho)}(f', f) + \lambda^* + d_{f,\mathcal{F}}^{(\rho)}(P, Q) \\
&\overset{(iv)}{\leq} \widetilde{\gamma}^* + \lambda^* + d_{f,\mathcal{F}}(P, Q) + \frac{2}{\rho}\widehat{\mathcal{R}}_S(\widetilde{\rho_{\mathcal{F}}\mathcal{F}}) + 6\sqrt{\frac{\log\frac{4}{\alpha}}{2n}}
\end{aligned}
$$

where $(i)$ is because the adversarial margin loss is larger than the standard margin loss; $(ii)$ is from Lemma A.4; $(iii)$ is from Lemma A.5; $(iv)$ is from (11) and holds with probability at least $1 - \alpha$. $\quad\square$

*Remark* A.7. Based on Theorem A.6, the following standard error bound for gradual AST can be derived similarly to Corollary 4.6. With the notations defined in Corollary 4.6, for any $\alpha \geq 0$, the following holds with the probability at least $1 - \alpha$ over the unlabeled data points $\{S_t\}_{t=1}^{T}$,

$$R_{P_T}(h_{f_T}) \leq R_{P_0}^{(\rho)}(f_0) + \sum_{t=1}^{T}\kappa_t + \frac{2T}{\rho}\widehat{\mathcal{R}}_S(\widetilde{\rho_{\mathcal{F}}\mathcal{F}}) + 6T\sqrt{\frac{\log\frac{4T}{\alpha}}{2n}}.$$

The result shows that the standard risk on target domain can be controlled by the standard margin loss on source domain and the discrepancy [53] between the intermediate domains.

## A.5 Proofs of Theorem 4.9

**Theorem 4.9.** *Given two distributions $P$ and $P_\eta$ over $\mathcal{X} \times \mathcal{Y}$ with the corresponding probability mass vectors $p$ and $p_\eta$. The following bound holds for any classifier $h \in \mathcal{H}_{\mathcal{F}}$.*

$$R_P(h) \leq \widetilde{R}_{P_\eta}(h) + \Psi^*(p, p_\eta, e, \Pi_\epsilon),$$

*where $e$ is the risk vector $e_i = \mathbb{1}[h(x_i) \neq y_i]$, and $\Pi_\epsilon = \{i : \exists\, \|\delta_i\|_q \leq \epsilon, h(x_i + \delta_i) \neq h(x_i)\}$.*

*Proof of Theorem 4.9.* Recall that the standard risk $R_P(h)$ on real data distribution is defined as follows:

$$R_P(h) = \sum_{(x,y)\in\mathcal{X}\times\mathcal{Y}} p(x, y)\mathbb{1}[h(x) \neq y],$$

and the adversarial risk $\widetilde{R}_{P_\eta}(h)$ on noisy-data distribution is defined as follows:

$$\widetilde{R}_{P_\eta}(h) = \sum_{(x,y)\in\mathcal{X}\times\mathcal{Y}} p_\eta(x,y) \max_{\|\delta\|_q\leq\epsilon} \mathbb{1}[h(x+\delta)\neq y].$$

We then have:

$$R_P(h) - \widetilde{R}_{P_\eta}(h) \leq \sum_{(x,y)\in\mathcal{X}\times\mathcal{Y}} p(x,y)\mathbb{1}[h(x)\neq y] - \sum_{(x,y)\in\mathcal{X}\times\mathcal{Y}} p_\eta(x,y) \max_{\|\delta\|_q\leq\epsilon} \mathbb{1}[h(x+\delta)\neq y]$$

$$= \sum_{(x,y)\in\mathcal{X}\times\mathcal{Y}} \left[ p(x,y)\mathbb{1}[h(x)\neq y] - p_\eta(x,y) \max_{\|\delta\|_q\leq\epsilon} \mathbb{1}[h(x+\delta)\neq y] \right]$$

$$= \min_{\|\delta\|_q\leq\epsilon} \sum_{(x,y)\in\mathcal{X}\times\mathcal{Y}} \left[ p(x,y)\mathbb{1}[h(x)\neq y] - p_\eta(x,y)\mathbb{1}[h(x+\delta)\neq y] \right]$$

We denote the risk vector as $e_i = \mathbb{1}[h(x_i)\neq y_i]$ and denote the adversarial risk vector as $\widetilde{e}_i = \mathbb{1}[h(x_i+\delta)\neq y_i]$. By the definition of $\Pi_\epsilon$:

$$\Pi_\epsilon = \{i : \exists \|\delta_i\|_q \leq \epsilon, h(x_i+\delta_i)\neq h(x_i)\},$$

we can get that $\widetilde{e}_i \neq e_i$ only if $i \in \Pi_\epsilon$. Then, we obtain that

$$R_P(h) - \widetilde{R}_{P_\eta}(h) \leq \min_{\widetilde{e}\in\{0,1\}^N} \left| p_\eta{}^T \widetilde{e} - p^T e \right|, \quad s.t. \ \widetilde{e}_i = e_i, \forall i \in \{1,\dots,N\}\backslash\Pi$$

$$= \Psi^*(p, p_\eta, e, \Pi_\epsilon).$$

$\square$

## A.6 Proofs of Theorem 4.11

**Theorem 4.11.** *For the real data distribution $P$ defined in Example 4.10, the optimal classifier that minimizes the standard risk $R_P(h)$ and adversarial risk $\widetilde{R}_P(h)$ is $h_0(\cdot) = sgn(\cdot)$. For the pseudo-labeled distribution $P_\eta$ defined in Example 4.10, the standardly trained classifier $h_{std}$ that minimizes the standard risk $R_{P_\eta}(h)$ is $h_{std} = h_w$; the adversarially trained classifier $h_{adv}$ that minimizes the adversarial risk $\widetilde{R}_{P_\eta}(h)$ has the following corresponding threshold $b_{adv}$,*

$$b_{adv} = \begin{cases} w + \epsilon, & \text{if } w < -\epsilon \\ 0, & \text{if } -\epsilon < w < \epsilon \\ w - \epsilon, & \text{if } w > \epsilon \end{cases}$$

Before we provide the proof, we first prove a lemma about the convex function.

**Lemma A.8.** *Let $t(x)$ be a convex function over $\mathbb{R}$. Given four points $a_1, a_2, a_3, a_4 \in \mathbb{R}$ such that $a_1 < a_2 < a_3 < a_4$ and $a_1 + a_4 = a_2 + a_3$, then the following holds:*

$$t(a_1) + t(a_4) \geq t(a_2) + t(a_3).$$

*Proof of Lemma A.8.* Let $\bar{a} \triangleq \frac{a_1+a_4}{2} = \frac{a_2+a_3}{2}$, $r \triangleq a_2 - a_1$ and $s \triangleq \bar{a} - a_2$.

Since $a_1 + a_4 = a_2 + a_3$, we then have $a_4 - a_3 = a_2 - a_1 = r$ and $a_3 - \bar{a} = a_3 - \frac{a_2+a_3}{2} = \frac{a_3-a_2}{2} = \frac{a_2+a_3}{2} - a_2 = s$. It is easy to see that:

$$a_2 = \frac{2s}{r+2s}a_1 + \frac{r}{r+2s}a_3 \overset{(i)}{\Rightarrow} t(a_2) \leq \frac{2s}{r+2s}t(a_1) + \frac{r}{r+2s}t(a_3),$$

$$a_3 = \frac{2s}{r+2s}a_4 + \frac{r}{r+2s}a_2 \overset{(ii)}{\Rightarrow} t(a_3) \leq \frac{2s}{r+2s}t(a_4) + \frac{r}{r+2s}t(a_2),$$

where $(i)$ and $(ii)$ are from the definition of convexity [7].

By combining these two inequalities, we have

$$t(a_2) + t(a_3) \leq \frac{2s}{r+2s}\left( t(a_1) + t(a_4) \right) + \frac{r}{r+2s}\left( t(a_3) + t(a_2) \right).$$

By rearranging the terms, we obtain the result

$$t(a_1) + t(a_4) \geq t(a_2) + t(a_3).$$

$\square$

***Proof of Theorem 4.11***. **(I). We first prove the results for real data distribution $P$.**

For the real data distribution $P$ defined in Example 4.10, the standard risk of a classifier $h_b \in \mathcal{H}$ can be written as follows:

$$R_P(h_b) = \frac{1}{2} \mathop{\mathbb{E}}_{x \sim P_{X|Y=1}} \mathbb{1}[h_b(x) \neq 1] + \frac{1}{2} \mathop{\mathbb{E}}_{x \sim P_{X|Y=-1}} \mathbb{1}[h_b(x) \neq -1].$$

By the definition of the classifier $h_b$, we have:

$$R_P(h_b) = \frac{1}{2} \mathop{\mathbb{E}}_{x \sim P_{X|Y=1}} \mathbb{1}[x \leq b] + \frac{1}{2} \mathop{\mathbb{E}}_{x \sim P_{X|Y=-1}} \mathbb{1}[x \geq b] = \frac{1}{2}\Phi_+(b) + \frac{1}{2}(1 - \Phi_-(b))$$

where $\Phi_+(b) = \frac{1}{\sigma\sqrt{2\pi}} \int_{-\infty}^{b} \exp\{-\frac{(x-\mu)^2}{2\sigma^2}\}dx$ is the cumulative distribution function (CDF) of the Gaussian distribution $\mathcal{N}(\mu, \sigma^2)$ and $\Phi_-(b) = \frac{1}{\sigma\sqrt{2\pi}} \int_{-\infty}^{b} \exp\{-\frac{(x+\mu)^2}{2\sigma^2}\}dx$ is the CDF of the Gaussian distribution $\mathcal{N}(-\mu, \sigma^2)$. By the symmetry of Gaussian distribution, we then have:

$$R_P(h_b) = \frac{1}{2}\Phi_+(b) + \frac{1}{2}\Phi_-(-2\mu - b).$$

The standard risk can be viewed as a function of $b$. By the rule of derivation for composition function, the derivative of the standard risk is

$$R_P'(h_b) = \frac{1}{2\sigma\sqrt{2\pi}}\exp\left\{-\frac{1}{2}\left(\frac{b-\mu}{\sigma}\right)^2\right\} - \frac{1}{2\sigma\sqrt{2\pi}}\exp\left\{-\frac{1}{2}\left(\frac{b+\mu}{\sigma}\right)^2\right\}$$

When $b > 0$, we can find that the derivative $R_P'(h_b)$ is larger than 0:

$$b > 0 \Rightarrow \left(\frac{b-\mu}{\sigma}\right)^2 < \left(\frac{b+\mu}{\sigma}\right)^2$$

$$\Rightarrow \exp\left\{-\frac{1}{2}\left(\frac{b-\mu}{\sigma}\right)^2\right\} > \exp\left\{-\frac{1}{2}\left(\frac{b+\mu}{\sigma}\right)^2\right\} \Rightarrow R_P'(h_b) > 0.$$

When $b < 0$, we can find that the derivative $R_P'(h_b)$ is smaller than 0:

$$b < 0 \Rightarrow \left(\frac{b-\mu}{\sigma}\right)^2 > \left(\frac{b+\mu}{\sigma}\right)^2$$

$$\Rightarrow \exp\left\{-\frac{1}{2}\left(\frac{b-\mu}{\sigma}\right)^2\right\} < \exp\left\{-\frac{1}{2}\left(\frac{b+\mu}{\sigma}\right)^2\right\} \Rightarrow R_P'(h_b) < 0.$$

When $b = 0$, it is easy to see that $R_P'(h_b) = 0$. Then we conclude that the standard risk $R_P(h_b)$ of the classifier $h_b$ achieves the minimal at $b = 0$ and the optimal classifier that minimizes the standard risk $R_P(h)$ is $h_0(\cdot) = sgn(\cdot)$.

The adversarial risk of a classifier $h_b \in \mathcal{H}$ over the real data distribution $P$ can be written as follows:

$$\widetilde{R}_P(h_b) = \frac{1}{2} \mathop{\mathbb{E}}_{x \sim P_{X|Y=1}} \mathbb{1}\left[\exists \delta, |\delta| \leq \epsilon, h_b(x + \delta) \neq 1\right] + \frac{1}{2} \mathop{\mathbb{E}}_{x \sim P_{X|Y=-1}} \mathbb{1}\left[\exists \delta, |\delta| \leq \epsilon, h_b(x + \delta) \neq -1\right].$$

By the definition of the classifier $h_b$, we have:

$$\widetilde{R}_P(h_b) = \frac{1}{2} \mathop{\mathbb{E}}_{x \sim P_{X|Y=1}} \mathbb{1}[x \leq b+\epsilon] + \frac{1}{2} \mathop{\mathbb{E}}_{x \sim P_{X|Y=-1}} \mathbb{1}[x \geq b-\epsilon] = \frac{1}{2}\Phi_+(b+\epsilon) + \frac{1}{2}(1 - \Phi_-(b-\epsilon)).$$

By the symmetry of Gaussian distribution, we then have:

$$\widetilde{R}_P(h_b) = \frac{1}{2}\Phi_+(b + \epsilon) + \frac{1}{2}\Phi_-(-2\mu - b + \epsilon).$$

By the rule of derivation for composition function, the derivative of the adversarial risk is

$$\widetilde{R}_P'(h_b) = \frac{1}{2\sigma\sqrt{2\pi}}\exp\left\{-\frac{1}{2}\left(\frac{b+\epsilon-\mu}{\sigma}\right)^2\right\} - \frac{1}{2\sigma\sqrt{2\pi}}\exp\left\{-\frac{1}{2}\left(\frac{b+\mu-\epsilon}{\sigma}\right)^2\right\}$$

Note that we have the assumption that $\epsilon \ll \mu$, which means $\epsilon - \mu < 0$. Similarly, when $b > 0$, we can find that $\widetilde{R}'_P(h_b) > 0$; when $b < 0$, we can find that $\widetilde{R}'_P(h_b) < 0$; when $b = 0$, $\widetilde{R}'_P(h_b) = 0$. Hence, the adversarial risk of the classifier $h_b$ over the real data distribution $P$ achieves minimal at the point $b = 0$. We conclude that the optimal classifier that minimizes the adversarial risk for the real data distribution is also $h_0(\cdot) = sgn(\cdot)$.

**(II). We next prove the results for the pseudo-labeled distribution $P_\eta$.**

It easy to see that the function $h_b$ achieves zero standard risk over the pseudo-labeled distribution $P_\eta$ if and only if $b = w$. Based on the assumption that the learner is provided with infinite samples, we can conclude that the standardly trained classifier $h_{std}$ that minimizes the standard risk is $h_{std} = h_w$.

For the adversarially trained classifier $h_{adv}$ that minimizes the adversarial risk $\widetilde{R}_{P_\eta}(h_b)$ over the pseudo-labeled distribution $P_\eta$, we first prove that the corresponding threshold satisfies:

$$b_{adv} \in [w - \epsilon, w + \epsilon].$$

The adversarial risk $\widetilde{R}_{P_\eta}(h_b)$ over the pseudo-labeled distribution $P_\eta$ can be written as follows:

$$\widetilde{R}_{P_\eta}(h_b) = \mathop{\mathbb{E}}_{(x,y) \sim P_\eta} \mathbb{1}\left[\exists \delta, |\delta| \leq \epsilon, h_b(x + \delta) \neq y\right].$$

We denote the marginal distribution over $\mathcal{X}$ of $P_\eta$ as $P_\eta^X$. Then we have:

$$\widetilde{R}_{P_\eta}(h_b) = \mathop{\mathbb{E}}_{x \sim P_\eta^X} \mathbb{1}\left[b - \epsilon < x \cap x < w\right] + \mathop{\mathbb{E}}_{x \sim P_\eta^X} \mathbb{1}[w < x \cap x < b + \epsilon], \tag{12}$$

where $\cap$ is the AND operation in logistical expressions.

When $b \leq w - \epsilon$, the second term in Eq. (12) equals $0$ since there does not exist an $x$ such that $w < x < b + \epsilon$. Then we derive that

$$\begin{aligned}
\widetilde{R}_{P_\eta}(h_b) &= \mathop{\mathbb{E}}_{x \sim P_\eta^X} \mathbb{1}\left[b - \epsilon < x \cap x < w\right] \\
&= \mathop{\mathbb{E}}_{x \sim P_\eta^X} \mathbb{1}\left[b - \epsilon < x < w\right] \\
&= \frac{1}{2}\left\{\Phi_-(w) - \Phi_-(b - \epsilon) + \Phi_+(w) - \Phi_+(b - \epsilon)\right\}
\end{aligned}$$

By the rule of derivation for composition function, the derivative of the adversarial risk is

$$\widetilde{R}'_{P_\eta}(h_b) = -\frac{1}{2\sigma\sqrt{2\pi}}\exp\left\{-\frac{1}{2}\left(\frac{b - \epsilon + \mu}{\sigma}\right)^2\right\} - \frac{1}{2\sigma\sqrt{2\pi}}\exp\left\{-\frac{1}{2}\left(\frac{b - \epsilon - \mu}{\sigma}\right)^2\right\}$$

Since the exponential function is positive, we have $\widetilde{R}'_{P_\eta}(h_b) < 0$. The adversarial risk over the pseudo-labeled distribution is a decreasing function on $(-\infty, w - \epsilon]$ and the adversarial risk achieves minimal at point $b = w - \epsilon$ when $b \leq w - \epsilon$.

On the other hand, when $b \geq w + \epsilon$, the first term in Eq. (12) equals $0$ since there does not exist an $x$ such that $b < x < w + \epsilon$. Then we derive that

$$\begin{aligned}
\widetilde{R}_{P_\eta}(h_b) &= \mathop{\mathbb{E}}_{x \sim P_\eta^X} \mathbb{1}\left[w < x \cap x < b + \epsilon\right] \\
&= \mathop{\mathbb{E}}_{x \sim P_\eta^X} \mathbb{1}\left[w < x < b + \epsilon\right] \\
&= \frac{1}{2}\left\{\Phi_-(b + \epsilon) - \Phi_-(w) + \Phi_+(b + \epsilon) - \Phi_+(w)\right\}
\end{aligned}$$

By the rule of derivation for composition function, the derivative of the adversarial risk is

$$\widetilde{R}'_{P_\eta}(h_b) = \frac{1}{2\sigma\sqrt{2\pi}}\exp\left\{-\frac{1}{2}\left(\frac{b + \epsilon + \mu}{\sigma}\right)^2\right\} + \frac{1}{2\sigma\sqrt{2\pi}}\exp\left\{-\frac{1}{2}\left(\frac{b + \epsilon - \mu}{\sigma}\right)^2\right\}$$

Since the exponential function is positive, we have $\widetilde{R}'_{P_\eta}(h_b) > 0$. The adversarial risk over the pseudo-labeled distribution is an increasing function on $[w+\epsilon, +\infty)$ and the adversarial risk achieves minimal at point $b = w + \epsilon$ when $b \geq w + \epsilon$.

By these two cases, we conclude that $b_{adv} \in [w - \epsilon, w + \epsilon]$.

Now, we focus on the case where $b \in [w - \epsilon, w + \epsilon]$. Then the adversarial risk can be written as follows:

$$\widetilde{R}_{P_\eta}(h_b) = \mathop{\mathbb{E}}_{x \sim P_\eta^X} \mathbb{1}\left[b - \epsilon < x < w\right] + \mathop{\mathbb{E}}_{x \sim P_\eta^X} \mathbb{1}\left[w < x < b + \epsilon\right] = \mathop{\mathbb{E}}_{x \sim P_\eta^X} \mathbb{1}\left[b - \epsilon < x < b + \epsilon\right].$$

By the definition of the Gaussian distribution, we have

$$\widetilde{R}_{P_\eta}(h_b) = \frac{1}{2} \left\{\Phi_-(b + \epsilon) - \Phi_-(b - \epsilon) + \Phi_+(b + \epsilon) - \Phi_+(b - \epsilon)\right\}.$$

Similarly, the derivative of the adversarial risk is

$$
\begin{aligned}
\widetilde{R}'_{P_\eta}(h_b) &= \frac{1}{2\sigma\sqrt{2\pi}} \exp\left\{-\frac{1}{2}\left(\frac{b + \epsilon + \mu}{\sigma}\right)^2\right\} - \frac{1}{2\sigma\sqrt{2\pi}} \exp\left\{-\frac{1}{2}\left(\frac{b - \epsilon + \mu}{\sigma}\right)^2\right\} \\
&\quad + \frac{1}{2\sigma\sqrt{2\pi}} \exp\left\{-\frac{1}{2}\left(\frac{b + \epsilon - \mu}{\sigma}\right)^2\right\} - \frac{1}{2\sigma\sqrt{2\pi}} \exp\left\{-\frac{1}{2}\left(\frac{b - \epsilon - \mu}{\sigma}\right)^2\right\} \\
&= \frac{1}{2\sigma\sqrt{2\pi}} \exp\left\{-\left(\frac{\mu + b + \epsilon}{\sqrt{2}\sigma}\right)^2\right\} - \frac{1}{2\sigma\sqrt{2\pi}} \exp\left\{-\left(\frac{\mu + b - \epsilon}{\sqrt{2}\sigma}\right)^2\right\} \\
&\quad + \frac{1}{2\sigma\sqrt{2\pi}} \exp\left\{-\left(\frac{\mu - b - \epsilon}{\sqrt{2}\sigma}\right)^2\right\} - \frac{1}{2\sigma\sqrt{2\pi}} \exp\left\{-\left(\frac{\mu - b + \epsilon}{\sqrt{2}\sigma}\right)^2\right\}
\end{aligned}
\tag{13}
$$

Let $\varphi(a) = e^{-a^2}$ be a function defined over $\mathbb{R}$. The first-order derivative of $\varphi(a)$ is $\varphi'(a) = -2ae^{-a^2}$. And the second-order derivative of $\varphi(a)$ is $\varphi''(a) = 2e^{-a^2}(\sqrt{2}a + 1)(\sqrt{2}a - 1)$. By the second-order derivative condition for convexity [7], we have

- When $a < -\frac{\sqrt{2}}{2}$ or $a > \frac{\sqrt{2}}{2}$, $\varphi''(a) > 0$. The function $\varphi(a)$ is convex on the interval $(-\infty, -\frac{\sqrt{2}}{2}) \cup (\frac{\sqrt{2}}{2}, +\infty)$.

- When $-\frac{\sqrt{2}}{2} < a < \frac{\sqrt{2}}{2}$, $\varphi''(a) < 0$. The function $\varphi(a)$ is concave on the interval $(-\frac{\sqrt{2}}{2}, \frac{\sqrt{2}}{2})$.

To simplify our notations, we use four points to denote the four terms in (13):

$$a_1 = \frac{\mu + b + \epsilon}{\sqrt{2}\sigma}, a_2 = \frac{\mu + b - \epsilon}{\sqrt{2}\sigma}, a_3 = \frac{\mu - b + \epsilon}{\sqrt{2}\sigma}, a_4 = \frac{\mu - b - \epsilon}{\sqrt{2}\sigma}.$$

Next we discuss the value of b in two cases.

If $b > 0$: By the assumption that $\mu$ is much larger than $b, \epsilon$ and $\mu$ is also larger than $\sigma$, we have:

$$\frac{\sqrt{2}}{2} < a_4 < a_3 < a_1, \frac{\sqrt{2}}{2} < a_4 < a_2 < a_1 \text{ and } a_1 + a_4 = a_2 + a_3.$$

By applying the Lemma A.8, we can obtain that $\widetilde{R}'_{P_\eta}(h_b) > 0$, which means $\widetilde{R}'_{P_\eta}(h_b)$ is an increasing function of $b$ when $b > 0$.

If $b < 0$: By the assumption that $\mu$ is much larger than $b, \epsilon$ and $\mu$ is also larger than $\sigma$, we have:

$$\frac{\sqrt{2}}{2} < a_2 < a_1 < a_3, \frac{\sqrt{2}}{2} < a_2 < a_4 < a_3 \text{ and } a_1 + a_4 = a_2 + a_3.$$

By applying the Lemma A.8, we can obtain that $\widetilde{R}'_{P_\eta}(h_b) < 0$, which means $\widetilde{R}'_{P_\eta}(h_b)$ is a decreasing function of $b$ when $b < 0$.

Based on the analysis, we conclude that:

Table 1: The performance (%) of gradual AST method on **MNIST** with various perturbation bound ($\epsilon$). The results in parentheses represent the improvement compared to the vanilla gradual self-training method. We use a 3-layer convolutional neural network.

| $\epsilon$ | 0.15 | 0.2 | 0.3 |
|---|---|---|---|
| $\mathcal{A}_{cle}$ | 97.11 (+8.17) | 96.65 (+7.71) | 96.59 (+8.65) |
| $\mathcal{A}_{adv}$ | 88.07 (+87.21) | 83.90 (+83.58) | 83.47 (+83.15) |

Table 2: The performance (%) of gradual AST method on **portraits** with various perturbation bound ($\epsilon$). The results in parentheses represent the improvement compared to the vanilla gradual self-training method. We use a 3-layer convolutional neural network.

| $\epsilon$ | 0.01 | 0.02 | 0.04 | 0.05 |
|---|---|---|---|---|
| $\mathcal{A}_{cle}$ | 83.89 (+2.50) | 85.19 (+3.80) | 84.16 (+2.77) | 84.92 (+3.53) |
| $\mathcal{A}_{adv}$ | 79.90 (+9.01) | 77.94 (+19.80) | 75.65 (+52.94) | 71.63 (+61.86) |

- When $w \leq -\epsilon$, $\widetilde{R}_{P_\eta}(h_b)$ is a decreasing function of $b$ on the interval $[w - \epsilon, w + \epsilon]$. Hence the optimal threshold $b_{adv} = w + \epsilon$.

- When $-\epsilon < w \leq \epsilon$, $\widetilde{R}_{P_\eta}(h_b)$ is a decreasing function of $b$ on the interval $(w - \epsilon, 0]$ and is an increasing function of $b$ on the interval $(0, w + \epsilon]$. Hence the optimal threshold $b_{adv} = 0$.

- When $\epsilon < w$, $\widetilde{R}_{P_\eta}(h_b)$ is an increasing function of $b$ on the interval $[w - \epsilon, w + \epsilon]$. Hence the optimal threshold $b_{adv} = w - \epsilon$.

$$b_{adv} = \begin{cases} w + \epsilon, & \text{if } w < -\epsilon \\ 0, & \text{if } -\epsilon < w < \epsilon \\ w - \epsilon, & \text{if } w > \epsilon \end{cases}$$

As we analyzed in part (I), the optimal classifier that minimizes the standard risk and the adversarial risk over the real data distribution is $h_0(\cdot) = sgn(\cdot)$. Compared to the standard training, the adversarial training has an effect of moving the threshold close to the optimal threshold $b = 0$. $\square$

## B  Additional Experiments

### B.1  Results of Methods with Varying Perturbation Radius $\epsilon$

In this section, we provide more experimental results of the methods with varying perturbation radius $\epsilon$. For the Rotating MNIST dataset, the perturbation radius is chosen from $\{0.15, 0.2, 0.3\}$; for the Portraits dataset, the perturbation radius is chosen from $\{0.01, 0.02, 0.04, 0.05\}$. The results are shown in Tables 1 and 2. As we can see from the tables, the proposed gradual AST method can consistently improve clean accuracy and adversarial robustness using various values of $\epsilon$.

### B.2  Results of Methods with Varying Domain Numbers

We split the Rotating MNIST and Portraits datasets with different lengths of interval and conduct an ablation study. For MNIST, the numbers of intermediate domains are 24, 30, and 42, respectively; for portraits, the numbers of intermediate domains are 8, 10, and 14, respectively. The results can be found in Tables 3 and 4, which show that our gradual AST method is nonsensitive to the choice of the number of intermediate domains.

### B.3  Results of Methods on Varying Neural Networks

In this section, we use ResNet-18 and ResNet-50 [21] as the backbones to validate the effectiveness of our proposed method. The results are presented in Tables 5 and 6, which show that complex networks can further improve performance (both clean accuracy and adversarial robustness).

Table 3: The performance (%) of gradual AST method on **MNIST** with various intermediate domain numbers (num.). The results in parentheses represent the improvement compared to the vanilla gradual self-training method. We set perturbation bound $\epsilon = 0.1$ and use a 3-layer convolutional neural network.

| num. | 24 | 30 | 42 |
|---|---|---|---|
| $\mathcal{A}_{cle}$ | 97.09 (+10.27) | 97.40 (+6.65) | 97.31 (+6.79) |
| $\mathcal{A}_{adv}$ | 90.24 (+84.45) | 90.72 (+84.94) | 90.30 (+82.22) |

Table 4: The performance (%) of gradual AST method on **portraits** with various intermediate domain numbers (num.). The results in parentheses represent the improvement compared to the vanilla gradual self-training method. We set perturbation bound $\epsilon = 0.031$ and use a 3-layer convolutional neural network.

| num. | 8 | 10 | 14 |
|---|---|---|---|
| $\mathcal{A}_{cle}$ | 84.84 (+2.72) | 85.99 (+0.26) | 84.44 (+1.60) |
| $\mathcal{A}_{adv}$ | 76.43 (+35.45) | 75.05 (+32.43) | 76.16 (+33.59) |

## B.4 Results of Methods with Varying Filtration Ratios $\zeta$

In this section, we provide more experimental results of the methods with varying filtration ratios $\zeta$ and starting domains $\tau$. For both of the Rotating MNIST and Portraits datasets, we chose the filtration ratios $\zeta$ from $\{0, 0.01, 0.02, 0.05, 0.1, 0.2\}$. In order to show the specific values, we present the results in the form of tables.

For Rotating MNIST dataset, we present the results of models with $\zeta = 0.2, 0.1, 0.05, 0.02, 0.01, 0$, in Tables 7, 8, 9, 10, 11, 12, respectively. As the tables show, the optimal filtration ratio $\zeta$ for gradual self-training ($\tau = 22$) is 0.1, which is used in previous work [26]. However, the gradual AST methods prefer smaller $\zeta$ which enables more data to be included. The optimal filtration ratio $\zeta$ for gradual AST (with the best starting domain $\tau$) is 0.05. When we set $\zeta = 0.05$, the gradual AST method with $\tau = 9$ achieves clean accuracy of $97.15\%$ and adversarial accuracy of $90.44\%$. This phenomenon indicates that AT has stronger anti-noisy ability than standard training.

For Portraits dataset, we present the results of models with $\zeta = 0.2, 0.1, 0.05, 0.02, 0.01, 0$, in Tables 13, 14, 15, 16, 17, 18, respectively. From the tables, we can draw a similar conclusion that the gradual AST methods prefer smaller $\zeta$.

## B.5 Training with Labeled Intermediate Domains

In this section, we conduct experiments where the learner is provided with labeled intermediate data. Although the learner has access to the ground-truth intermediate labels, we still keep the filtering process, since we need to control the same data size for comparison. We present the results of the models with varying $\tau$ in Table 19 and Table 20. Since the results in Section 3.3 show that $\zeta = 0.05$ is the optimal filtration ratio for gradual AST methods, we set $\zeta = 0.05$ in this section. From Table 19, we can see that, if the learner is provided with labeled intermediate domains, the gradual self-training method ($\tau = 22$) achieves clean accuracy of $98.44\%$ on Rotating MNIST. Recall the results in Table 9 showing that the proposed gradual AST method ($\tau = 0$) achieves clean accuracy of $95.12\%$. The

Table 5: The performance (%) of gradual AST method on **MNIST** with various backbone networks. The results in parentheses represent the improvement compared to the vanilla gradual self-training method. We set perturbation bound $\epsilon = 0.1$.

| backbone | ResNet18 | ResNet50 |
|---|---|---|
| $\mathcal{A}_{cle}$ | 98.70 (+0.02) | 98.47 (+0.44) |
| $\mathcal{A}_{adv}$ | 96.57 (+89.11) | 96.13 (+22.05) |

Table 6: The performance (%) of gradual AST method on **portraits** with various backbone networks. The results in parentheses represent the improvement compared to the vanilla gradual self-training method. We set perturbation bound $\epsilon = 0.031$.

| backbone | ResNet18 | ResNet50 |
|----------|----------|----------|
| $\mathcal{A}_{cle}$ | 86.52 (+0.52) | 87.09 (+0.0) |
| $\mathcal{A}_{adv}$ | 78.12 (+52.08) | 79.01 (+47.24) |

performance (95.12%) of the proposed gradual AST is close to the optimal performance (98.44%) of the gradual self-training where the learner is provided with labeled intermediate domains.

## B.6 Visualization of the Filter

In order to better demonstrate the filtering effect, we use the t-distributed stochastic neighbor embedding (t-SNE) to visualize the data in each domain. t-SNE [13] is a statistical method for visualizing high-dimensional data by giving each data point a location in a two or three-dimensional map.

We present the t-SNE visualizations for each intermediate domain on Rotating MNIST in Figure 3-6. Each sub-figure represents a domain. We use red points to denote the correctly pseudo-labeled data predicted by the model. We use blue and green points to denote the data with incorrect pseudo-labels, where the blue points are successfully filtered by filtration and the green points are retained.

As we can see in the figures, the blue and green points are only a small part of the whole, which indicates most pseudo-labels of the intermediate data generated by the model are correct. Furthermore, the blue points make up a large portion of the non-red dots, which means the incorrect pseudo-labels are effectively filtered out.

Table 7: The results of methods with varying starting domains $\tau$ on **Rotating MNIST**. The filtration ratio $\zeta$ is set to **0.2**. We use $\mathcal{A}_{cle}$ to denote the clean accuracy (%) and use $\mathcal{A}_{adv}$ to denote the adversarial accuracy (%). Results in bold indicate the best performance.

| $\tau_0$ | $\mathcal{A}_{cle}$ | $\mathcal{A}_{adv}$ | $\epsilon$ | $\zeta$ |
|---|---|---|---|---|
| 0 | 90.23 | 81.06 | 0.1 | 0.2 |
| 1 | 93.15 | 82.59 | 0.1 | 0.2 |
| 2 | 93.67 | 83.96 | 0.1 | 0.2 |
| 3 | 92.59 | 81.94 | 0.1 | 0.2 |
| 4 | 93.97 | 83.50 | 0.1 | 0.2 |
| 5 | 94.78 | 85.00 | 0.1 | 0.2 |
| 6 | 92.14 | 82.47 | 0.1 | 0.2 |
| 7 | 94.51 | 84.77 | 0.1 | 0.2 |
| 8 | 94.18 | 83.32 | 0.1 | 0.2 |
| 9 | 91.37 | 81.35 | 0.1 | 0.2 |
| 10 | 95.00 | **85.14** | 0.1 | 0.2 |
| 11 | 94.57 | 84.15 | 0.1 | 0.2 |
| 12 | 95.08 | 84.82 | 0.1 | 0.2 |
| 13 | **95.10** | 83.84 | 0.1 | 0.2 |
| 14 | 94.82 | 84.17 | 0.1 | 0.2 |
| 15 | 91.42 | 80.89 | 0.1 | 0.2 |
| 16 | 92.26 | 81.64 | 0.1 | 0.2 |
| 17 | 91.68 | 80.49 | 0.1 | 0.2 |
| 18 | 91.48 | 79.85 | 0.1 | 0.2 |
| 19 | 91.76 | 79.63 | 0.1 | 0.2 |
| 20 | 90.13 | 77.00 | 0.1 | 0.2 |
| 21 | 88.52 | 73.44 | 0.1 | 0.2 |
| 22 | 87.42 | 8.73 | 0.1 | 0.2 |

Table 8: The results of methods with varying starting domains $\tau$ on **Rotating MNIST**. The filtration ratio $\zeta$ is set to **0.1**. We use $\mathcal{A}_{cle}$ to denote the clean accuracy (%) and use $\mathcal{A}_{adv}$ to denote the adversarial accuracy (%). Results in bold indicate the best performance.

| $\tau_0$ | $\mathcal{A}_{cle}$ | $\mathcal{A}_{adv}$ | $\epsilon$ | $\zeta$ |
|---|---|---|---|---|
| 0 | 90.59 | 83.54 | 0.1 | 0.1 |
| 1 | 95.65 | 88.03 | 0.1 | 0.1 |
| 2 | 95.52 | 88.02 | 0.1 | 0.1 |
| 3 | 95.28 | 87.11 | 0.1 | 0.1 |
| 4 | **96.66** | **89.16** | 0.1 | 0.1 |
| 5 | 96.09 | 88.13 | 0.1 | 0.1 |
| 6 | 95.98 | 88.70 | 0.1 | 0.1 |
| 7 | 96.58 | 89.04 | 0.1 | 0.1 |
| 8 | 96.01 | 88.85 | 0.1 | 0.1 |
| 9 | 96.66 | 88.86 | 0.1 | 0.1 |
| 10 | 96.65 | 88.48 | 0.1 | 0.1 |
| 11 | 96.82 | 88.79 | 0.1 | 0.1 |
| 12 | 96.10 | 88.25 | 0.1 | 0.1 |
| 13 | 96.51 | 88.27 | 0.1 | 0.1 |
| 14 | 95.59 | 86.65 | 0.1 | 0.1 |
| 15 | 95.70 | 86.55 | 0.1 | 0.1 |
| 16 | 95.00 | 85.70 | 0.1 | 0.1 |
| 17 | 94.92 | 85.02 | 0.1 | 0.1 |
| 18 | 94.25 | 84.44 | 0.1 | 0.1 |
| 19 | 93.82 | 83.14 | 0.1 | 0.1 |
| 20 | 91.50 | 80.19 | 0.1 | 0.1 |
| 21 | 91.05 | 76.50 | 0.1 | 0.1 |
| 22 | 90.06 | 6.00 | 0.1 | 0.1 |

Table 9: The results of methods with varying starting domains $\tau$ on **Rotating MNIST**. The filtration ratio $\zeta$ is set to **0.05**. We use $\mathcal{A}_{cle}$ to denote the clean accuracy (%) and use $\mathcal{A}_{adv}$ to denote the adversarial accuracy (%). Results in bold indicate the best performance.

| $\tau_0$ | $\mathcal{A}_{cle}$ | $\mathcal{A}_{adv}$ | $\epsilon$ | $\zeta$ |
|---|---|---|---|---|
| 0 | 95.12 | 89.00 | 0.1 | 0.05 |
| 1 | 96.30 | 90.14 | 0.1 | 0.05 |
| 2 | 95.54 | 88.70 | 0.1 | 0.05 |
| 3 | 96.41 | 89.53 | 0.1 | 0.05 |
| 4 | 95.96 | 89.73 | 0.1 | 0.05 |
| 5 | 96.58 | 89.82 | 0.1 | 0.05 |
| 6 | 96.53 | 89.65 | 0.1 | 0.05 |
| 7 | 96.89 | 90.06 | 0.1 | 0.05 |
| 8 | 96.99 | 89.63 | 0.1 | 0.05 |
| 9 | **97.14** | **90.44** | 0.1 | 0.05 |
| 10 | 96.88 | 89.62 | 0.1 | 0.05 |
| 11 | 95.76 | 88.06 | 0.1 | 0.05 |
| 12 | 95.52 | 87.94 | 0.1 | 0.05 |
| 13 | 95.72 | 88.47 | 0.1 | 0.05 |
| 14 | 95.50 | 87.27 | 0.1 | 0.05 |
| 15 | 93.41 | 84.81 | 0.1 | 0.05 |
| 16 | 92.60 | 83.96 | 0.1 | 0.05 |
| 17 | 92.48 | 83.84 | 0.1 | 0.05 |
| 18 | 92.77 | 83.00 | 0.1 | 0.05 |
| 19 | 89.24 | 78.77 | 0.1 | 0.05 |
| 20 | 90.33 | 79.44 | 0.1 | 0.05 |
| 21 | 89.93 | 76.39 | 0.1 | 0.05 |
| 22 | 88.96 | 7.01 | 0.1 | 0.05 |

Table 10: The results of methods with varying starting domains $\tau$ on **Rotating MNIST**. The filtration ratio $\zeta$ is set to **0.02**. We use $\mathcal{A}_{cle}$ to denote the clean accuracy (%) and use $\mathcal{A}_{adv}$ to denote the adversarial accuracy (%). Results in bold indicate the best performance.

| $\tau_0$ | $\mathcal{A}_{cle}$ | $\mathcal{A}_{adv}$ | $\epsilon$ | $\zeta$ |
|---|---|---|---|---|
| 0 | 96.05 | 90.20 | 0.1 | 0.02 |
| 1 | 96.31 | 90.01 | 0.1 | 0.02 |
| 2 | 96.13 | 89.47 | 0.1 | 0.02 |
| 3 | **96.87** | **90.31** | 0.1 | 0.02 |
| 4 | 95.32 | 88.67 | 0.1 | 0.02 |
| 5 | 96.50 | 90.25 | 0.1 | 0.02 |
| 6 | 96.79 | 90.04 | 0.1 | 0.02 |
| 7 | 96.41 | 89.53 | 0.1 | 0.02 |
| 8 | 96.67 | 89.52 | 0.1 | 0.02 |
| 9 | 96.77 | 89.82 | 0.1 | 0.02 |
| 10 | 95.53 | 88.26 | 0.1 | 0.02 |
| 11 | 96.18 | 88.71 | 0.1 | 0.02 |
| 12 | 91.10 | 83.05 | 0.1 | 0.02 |
| 13 | 90.52 | 83.36 | 0.1 | 0.02 |
| 14 | 90.00 | 82.27 | 0.1 | 0.02 |
| 15 | 91.34 | 82.78 | 0.1 | 0.02 |
| 16 | 89.61 | 80.45 | 0.1 | 0.02 |
| 17 | 87.60 | 78.39 | 0.1 | 0.02 |
| 18 | 85.88 | 76.03 | 0.1 | 0.02 |
| 19 | 85.88 | 75.15 | 0.1 | 0.02 |
| 20 | 84.35 | 72.23 | 0.1 | 0.02 |
| 21 | 83.83 | 69.12 | 0.1 | 0.02 |
| 22 | 82.43 | 6.44 | 0.1 | 0.02 |

Table 11: The results of methods with varying starting domains $\tau$ on **Rotating MNIST**. The filtration ratio $\zeta$ is set to **0.01**. We use $\mathcal{A}_{cle}$ to denote the clean accuracy (%) and use $\mathcal{A}_{adv}$ to denote the adversarial accuracy (%). Results in bold indicate the best performance.

| $\tau_0$ | $\mathcal{A}_{cle}$ | $\mathcal{A}_{adv}$ | $\epsilon$ | $\zeta$ |
|---|---|---|---|---|
| 0 | 95.20 | 89.55 | 0.1 | 0.01 |
| 1 | 96.14 | 89.88 | 0.1 | 0.01 |
| 2 | 96.00 | 89.74 | 0.1 | 0.01 |
| 3 | 96.60 | 89.82 | 0.1 | 0.01 |
| 4 | 96.29 | 90.02 | 0.1 | 0.01 |
| 5 | 96.82 | **90.49** | 0.1 | 0.01 |
| 6 | 96.69 | 90.11 | 0.1 | 0.01 |
| 7 | 96.53 | 89.69 | 0.1 | 0.01 |
| 8 | 96.74 | 90.31 | 0.1 | 0.01 |
| 9 | **96.83** | 89.70 | 0.1 | 0.01 |
| 10 | 96.37 | 88.71 | 0.1 | 0.01 |
| 11 | 95.42 | 87.79 | 0.1 | 0.01 |
| 12 | 94.43 | 86.20 | 0.1 | 0.01 |
| 13 | 95.42 | 87.32 | 0.1 | 0.01 |
| 14 | 94.25 | 85.95 | 0.1 | 0.01 |
| 15 | 93.31 | 85.18 | 0.1 | 0.01 |
| 16 | 93.08 | 84.44 | 0.1 | 0.01 |
| 17 | 92.74 | 83.49 | 0.1 | 0.01 |
| 18 | 92.45 | 82.66 | 0.1 | 0.01 |
| 19 | 91.85 | 80.71 | 0.1 | 0.01 |
| 20 | 91.47 | 79.47 | 0.1 | 0.01 |
| 21 | 90.25 | 75.84 | 0.1 | 0.01 |
| 22 | 88.57 | 6.49 | 0.1 | 0.01 |

Table 12: The results of methods with varying starting domains $\tau$ on **Rotating MNIST**. The filtration ratio $\zeta$ is set to **0**. We use $\mathcal{A}_{cle}$ to denote the clean accuracy (%) and use $\mathcal{A}_{adv}$ to denote the adversarial accuracy (%). Results in bold indicate the best performance.

| $\tau_0$ | $\mathcal{A}_{cle}$ | $\mathcal{A}_{adv}$ | $\epsilon$ | $\zeta$ |
|---|---|---|---|---|
| 0 | 94.86 | 89.26 | 0.1 | 0 |
| 1 | 95.48 | 90.47 | 0.1 | 0 |
| 2 | 95.67 | 89.69 | 0.1 | 0 |
| 3 | 95.96 | 89.95 | 0.1 | 0 |
| 4 | 96.58 | **90.90** | 0.1 | 0 |
| 5 | 96.60 | 90.74 | 0.1 | 0 |
| 6 | 95.84 | 88.69 | 0.1 | 0 |
| 7 | **96.87** | 90.16 | 0.1 | 0 |
| 8 | 95.97 | 89.75 | 0.1 | 0 |
| 9 | 92.35 | 85.78 | 0.1 | 0 |
| 10 | 96.66 | 90.05 | 0.1 | 0 |
| 11 | 92.50 | 85.46 | 0.1 | 0 |
| 12 | 94.86 | 87.91 | 0.1 | 0 |
| 13 | 94.08 | 86.41 | 0.1 | 0 |
| 14 | 91.03 | 83.52 | 0.1 | 0 |
| 15 | 90.63 | 82.80 | 0.1 | 0 |
| 16 | 91.16 | 82.61 | 0.1 | 0 |
| 17 | 88.64 | 79.38 | 0.1 | 0 |
| 18 | 90.34 | 80.50 | 0.1 | 0 |
| 19 | 89.94 | 79.30 | 0.1 | 0 |
| 20 | 87.99 | 76.25 | 0.1 | 0 |
| 21 | 86.90 | 72.24 | 0.1 | 0 |
| 22 | 84.88 | 3.97 | 0.1 | 0 |

Table 13: The results of methods with varying starting domains $\tau$ on **Portraits**. The filtration ratio $\zeta$ is set to **0.2**. We use $\mathcal{A}_{cle}$ to denote the clean accuracy (%) and use $\mathcal{A}_{adv}$ to denote the adversarial accuracy (%). Results in bold indicate the best performance.

| $\tau_0$ | $\mathcal{A}_{cle}$ | $\mathcal{A}_{adv}$ | $\epsilon$ | $\zeta$ |
|---|---|---|---|---|
| 0 | 84.09 | 74.53 | 0.031 | 0.2 |
| 1 | 82.94 | 73.83 | 0.031 | 0.2 |
| 2 | 84.62 | **75.44** | 0.031 | 0.2 |
| 3 | **84.65** | 74.95 | 0.031 | 0.2 |
| 4 | 82.80 | 72.59 | 0.031 | 0.2 |
| 5 | 81.51 | 69.10 | 0.031 | 0.2 |
| 6 | 82.02 | 69.33 | 0.031 | 0.2 |
| 7 | 82.82 | 68.55 | 0.031 | 0.2 |
| 8 | 81.79 | 40.54 | 0.031 | 0.2 |

Table 14: The results of methods with varying starting domains $\tau$ on **Portraits**. The filtration ratio $\zeta$ is set to **0.1**. We use $\mathcal{A}_{cle}$ to denote the clean accuracy (%) and use $\mathcal{A}_{adv}$ to denote the adversarial accuracy (%). Results in bold indicate the best performance.

| $\tau_0$ | $\mathcal{A}_{cle}$ | $\mathcal{A}_{adv}$ | $\epsilon$ | $\zeta$ |
|---|---|---|---|---|
| 0 | 84.77 | **77.64** | 0.031 | 0.1 |
| 1 | 84.28 | 76.07 | 0.031 | 0.1 |
| 2 | **85.45** | 76.27 | 0.031 | 0.1 |
| 3 | 83.20 | 74.61 | 0.031 | 0.1 |
| 4 | 81.93 | 71.68 | 0.031 | 0.1 |
| 5 | 81.05 | 69.82 | 0.031 | 0.1 |
| 6 | 82.23 | 71.88 | 0.031 | 0.1 |
| 7 | 84.77 | 73.93 | 0.031 | 0.1 |
| 8 | 82.03 | 40.23 | 0.031 | 0.1 |

Table 15: The results of methods with varying starting domains $\tau$ on **Portraits**. The filtration ratio $\zeta$ is set to **0.05**. We use $\mathcal{A}_{cle}$ to denote the clean accuracy (%) and use $\mathcal{A}_{adv}$ to denote the adversarial accuracy (%). Results in bold indicate the best performance.

| $\tau_0$ | $\mathcal{A}_{cle}$ | $\mathcal{A}_{adv}$ | $\epsilon$ | $\zeta$ |
|---|---|---|---|---|
| 0 | 83.89 | 76.76 | 0.031 | 0.05 |
| 1 | **86.04** | 77.25 | 0.031 | 0.05 |
| 2 | 85.64 | **77.64** | 0.031 | 0.05 |
| 3 | 84.28 | 74.90 | 0.031 | 0.05 |
| 4 | 83.20 | 74.32 | 0.031 | 0.05 |
| 5 | 81.25 | 72.17 | 0.031 | 0.05 |
| 6 | 81.64 | 71.58 | 0.031 | 0.05 |
| 7 | 83.69 | 72.66 | 0.031 | 0.05 |
| 8 | 81.35 | 39.75 | 0.031 | 0.05 |

Table 16: The results of methods with varying starting domains $\tau$ on **Portraits**. The filtration ratio $\zeta$ is set to **0.02**. We use $\mathcal{A}_{cle}$ to denote the clean accuracy (%) and use $\mathcal{A}_{adv}$ to denote the adversarial accuracy (%). Results in bold indicate the best performance.

| $\tau_0$ | $\mathcal{A}_{cle}$ | $\mathcal{A}_{adv}$ | $\epsilon$ | $\zeta$ |
|---|---|---|---|---|
| 0 | 84.67 | **77.94** | 0.031 | 0.02 |
| 1 | 84.08 | 77.60 | 0.031 | 0.02 |
| 2 | 83.81 | 77.91 | 0.031 | 0.02 |
| 3 | 83.15 | 75.83 | 0.031 | 0.02 |
| 4 | 82.66 | 73.81 | 0.031 | 0.02 |
| 5 | 82.66 | 74.98 | 0.031 | 0.02 |
| 6 | 82.66 | 73.10 | 0.031 | 0.02 |
| 7 | **85.28** | 75.44 | 0.031 | 0.02 |
| 8 | 82.66 | 42.51 | 0.031 | 0.02 |

Table 17: The results of methods with varying starting domains $\tau$ on **Portraits**. The filtration ratio $\zeta$ is set to **0.01**. We use $\mathcal{A}_{cle}$ to denote the clean accuracy (%) and use $\mathcal{A}_{adv}$ to denote the adversarial accuracy (%). Results in bold indicate the best performance.

| $\tau_0$ | $\mathcal{A}_{cle}$ | $\mathcal{A}_{adv}$ | $\epsilon$ | $\zeta$ |
|---|---|---|---|---|
| 0 | 83.29 | 76.55 | 0.031 | 0.01 |
| 1 | 84.57 | **78.60** | 0.031 | 0.01 |
| 2 | 84.79 | 76.97 | 0.031 | 0.01 |
| 3 | **85.33** | 77.85 | 0.031 | 0.01 |
| 4 | 85.13 | 76.77 | 0.031 | 0.01 |
| 5 | 78.91 | 70.47 | 0.031 | 0.01 |
| 6 | 81.19 | 71.86 | 0.031 | 0.01 |
| 7 | 81.75 | 71.86 | 0.031 | 0.01 |
| 8 | 80.78 | 38.57 | 0.031 | 0.01 |

Table 18: The results of methods with varying starting domains $\tau$ on **Portraits**. The filtration ratio $\zeta$ is set to **0**. We use $\mathcal{A}_{cle}$ to denote the clean accuracy (%) and use $\mathcal{A}_{adv}$ to denote the adversarial accuracy (%). Results in bold indicate the best performance.

| $\tau_0$ | $\mathcal{A}_{cle}$ | $\mathcal{A}_{adv}$ | $\epsilon$ | $\zeta$ |
|---|---|---|---|---|
| 0 | 82.42 | 76.37 | 0.031 | 0 |
| 1 | 84.67 | 77.25 | 0.031 | 0 |
| 2 | 83.69 | 77.93 | 0.031 | 0 |
| 3 | 84.47 | 77.83 | 0.031 | 0 |
| 4 | 85.25 | **78.03** | 0.031 | 0 |
| 5 | 79.79 | 74.32 | 0.031 | 0 |
| 6 | 81.93 | 74.71 | 0.031 | 0 |
| 7 | **86.04** | 75.98 | 0.031 | 0 |
| 8 | 81.93 | 33.69 | 0.031 | 0 |

Table 19: The results of methods with varying starting domains $\tau$ on **Rotating MNIST**. **The learner is provided with labeled intermediate data.** The filtration ratio $\zeta$ is set to **0.05**. We use $\mathcal{A}_{cle}$ to denote the clean accuracy (%) and use $\mathcal{A}_{adv}$ to denote the adversarial accuracy (%). Results in bold indicate the best performance.

| $\tau_0$ | $\mathcal{A}_{cle}$ | $\mathcal{A}_{adv}$ | $\epsilon$ | $\zeta$ |
|---|---|---|---|---|
| 0 | 98.65 | **93.67** | 0.1 | 0.05 |
| 1 | 98.52 | 93.49 | 0.1 | 0.05 |
| 2 | 98.53 | 93.53 | 0.1 | 0.05 |
| 3 | 98.54 | 93.02 | 0.1 | 0.05 |
| 4 | 98.68 | 92.65 | 0.1 | 0.05 |
| 5 | 98.73 | 93.00 | 0.1 | 0.05 |
| 6 | 98.61 | 93.21 | 0.1 | 0.05 |
| 7 | 98.72 | 92.92 | 0.1 | 0.05 |
| 8 | 98.85 | 93.13 | 0.1 | 0.05 |
| 9 | 98.71 | 92.62 | 0.1 | 0.05 |
| 10 | 98.73 | 92.54 | 0.1 | 0.05 |
| 11 | 98.82 | 92.15 | 0.1 | 0.05 |
| 12 | 98.80 | 91.98 | 0.1 | 0.05 |
| 13 | **98.87** | 92.01 | 0.1 | 0.05 |
| 14 | 98.71 | 91.64 | 0.1 | 0.05 |
| 15 | 98.69 | 91.46 | 0.1 | 0.05 |
| 16 | 98.67 | 91.37 | 0.1 | 0.05 |
| 17 | 98.73 | 90.67 | 0.1 | 0.05 |
| 18 | 98.63 | 90.04 | 0.1 | 0.05 |
| 19 | 98.67 | 89.03 | 0.1 | 0.05 |
| 20 | 98.51 | 88.25 | 0.1 | 0.05 |
| 21 | 98.44 | 85.70 | 0.1 | 0.05 |
| 22 | 98.44 | 3.59 | 0.1 | 0.05 |

Table 20: The results of methods with varying starting domains $\tau$ on **Portraits**. **The learner is provided with labeled intermediate data.** The filtration ratio $\zeta$ is set to **0.05**. We use $\mathcal{A}_{cle}$ to denote the clean accuracy (%) and use $\mathcal{A}_{adv}$ to denote the adversarial accuracy (%). Results in bold indicate the best performance.

| $\tau_0$ | $\mathcal{A}_{cle}$ | $\mathcal{A}_{adv}$ | $\epsilon$ | $\zeta$ |
|---|---|---|---|---|
| 0 | 89.55 | 78.22 | 0.031 | 0.05 |
| 1 | 90.33 | 79.69 | 0.031 | 0.05 |
| 2 | 90.72 | 79.10 | 0.031 | 0.05 |
| 3 | 90.43 | 79.10 | 0.031 | 0.05 |
| 4 | 90.33 | 78.91 | 0.031 | 0.05 |
| 5 | 90.53 | **79.69** | 0.031 | 0.05 |
| 6 | 90.43 | 78.91 | 0.031 | 0.05 |
| 7 | 89.55 | 78.13 | 0.031 | 0.05 |
| 8 | **91.99** | 19.82 | 0.031 | 0.05 |

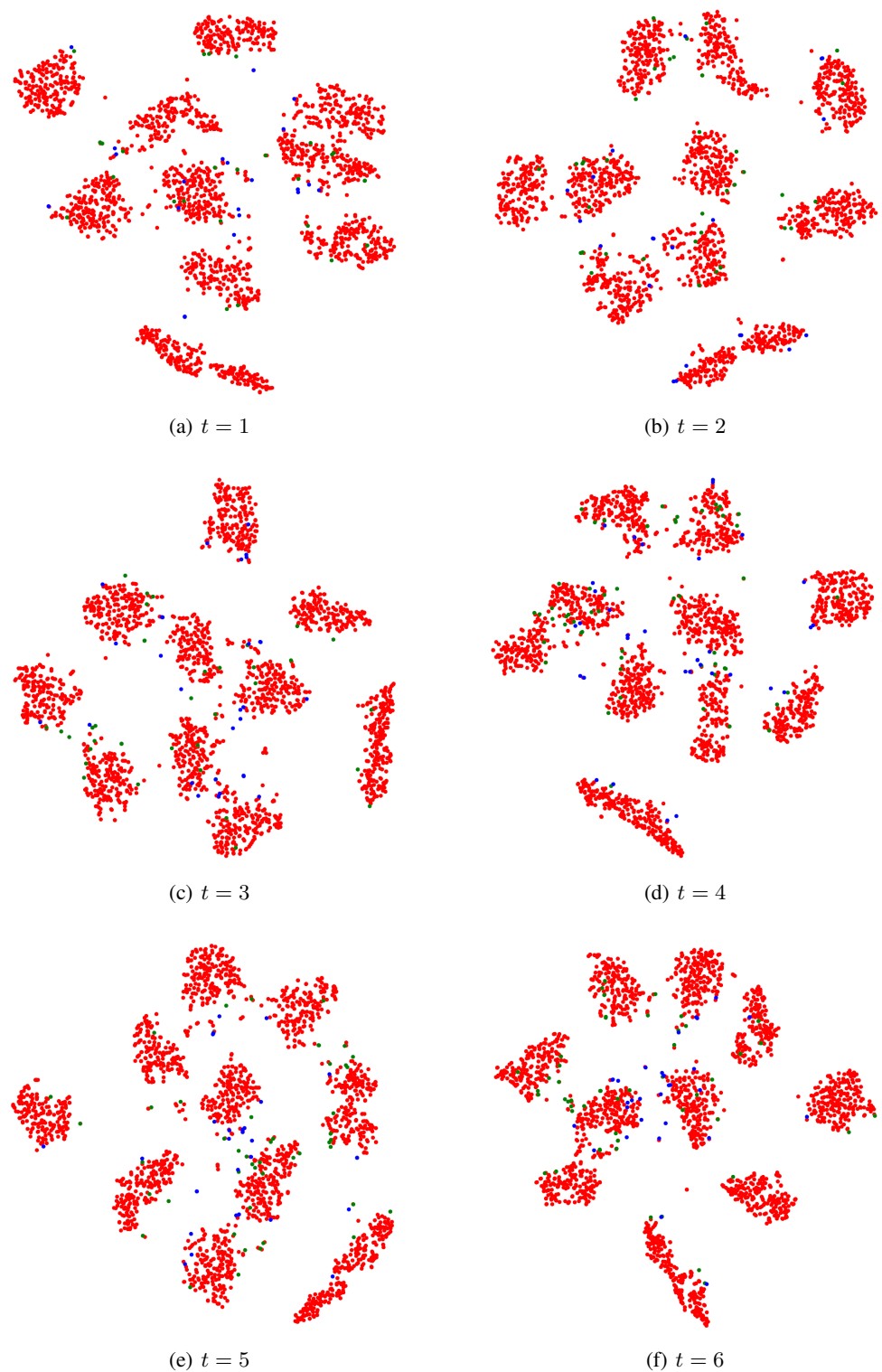

(a) $t = 1$

(b) $t = 2$

(c) $t = 3$

(d) $t = 4$

(e) $t = 5$

(f) $t = 6$

Figure 3: Visualization (I) of the data points in each domain using t-SNE. We use red points to denote the correctly pseudo-labeled data predicted by the model. We use blue and green points to denote the data with incorrect pseudo-labels, where the blue points are successfully filtered by the filter and the green points are retained. Best viewed in color.

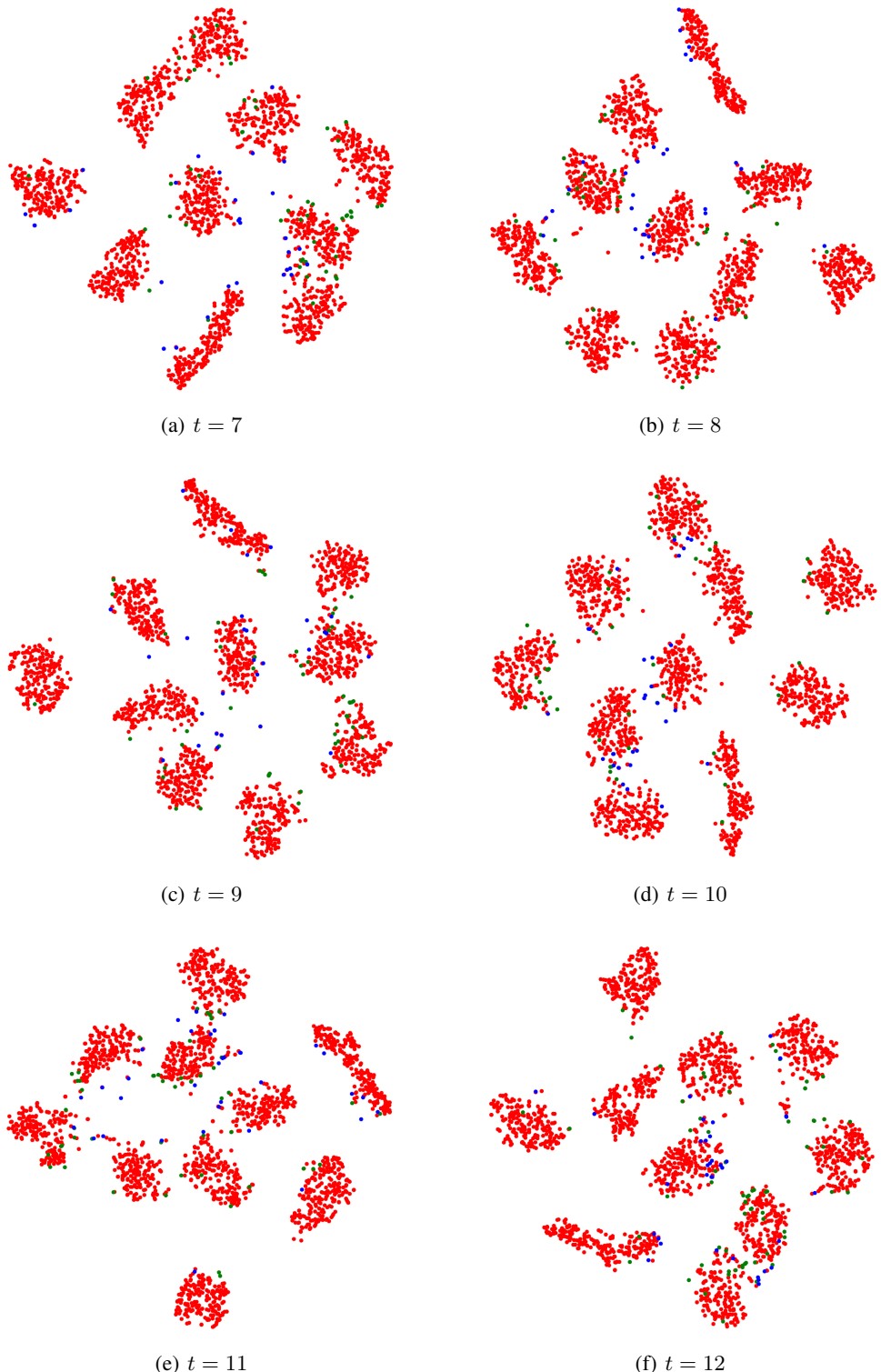

(a) $t = 7$  (b) $t = 8$

(c) $t = 9$  (d) $t = 10$

(e) $t = 11$  (f) $t = 12$

Figure 4: Visualization (II) of the data points in each domain using t-SNE. We use red points to denote the correctly pseudo-labeled data predicted by the model. We use blue and green points to denote the data with incorrect pseudo-labels, where the blue points are successfully filtered by the filter and the green points are retained. Best viewed in color.

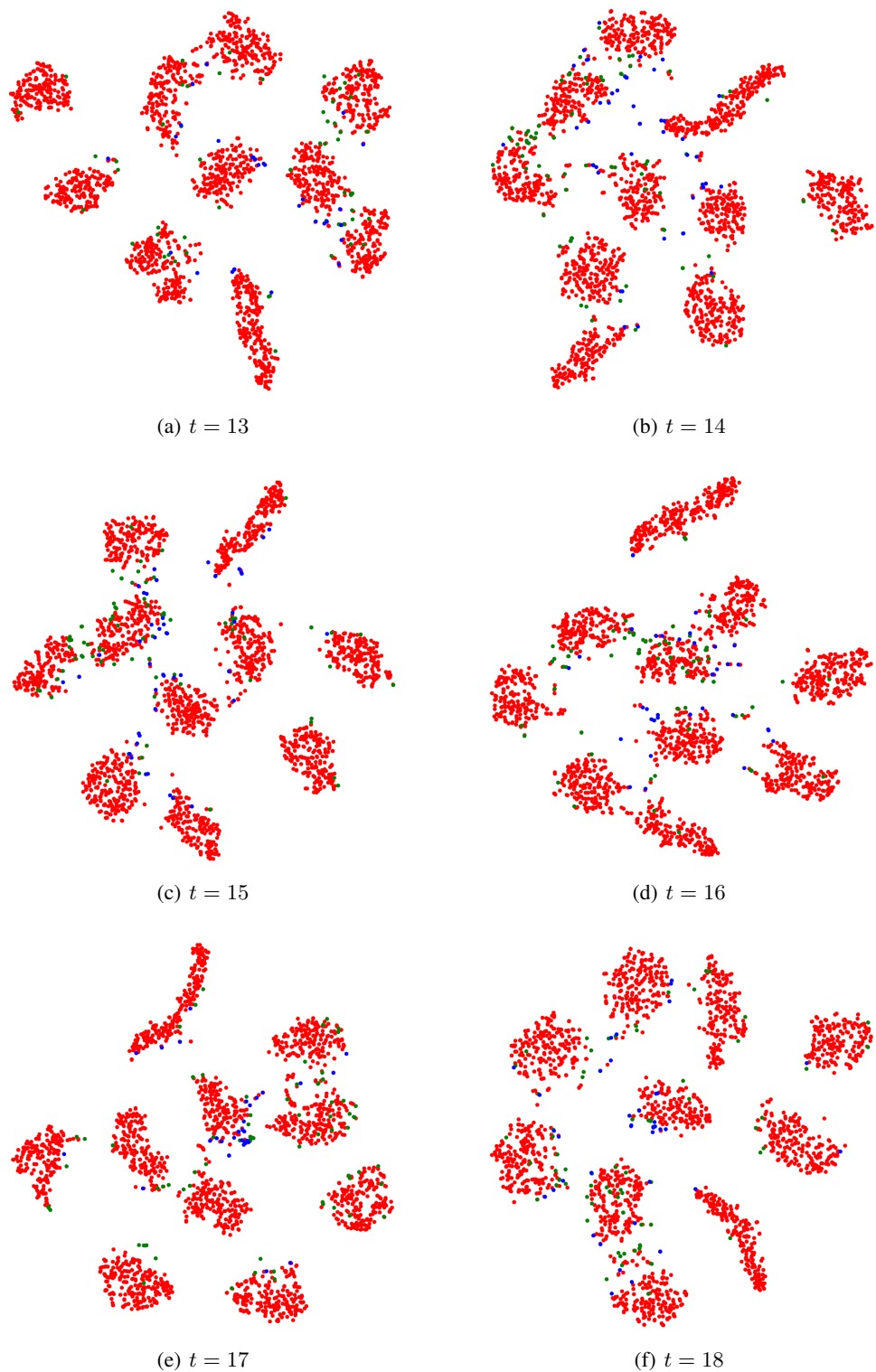

(a) $t = 13$

(b) $t = 14$

(c) $t = 15$

(d) $t = 16$

(e) $t = 17$

(f) $t = 18$

Figure 5: Visualization (III) of the data points in each domain using t-SNE. We use red points to denote the correctly pseudo-labeled data predicted by the model. We use blue and green points to denote the data with incorrect pseudo-labels, where the blue points are successfully filtered by the filter and the green points are retained. Best viewed in color.

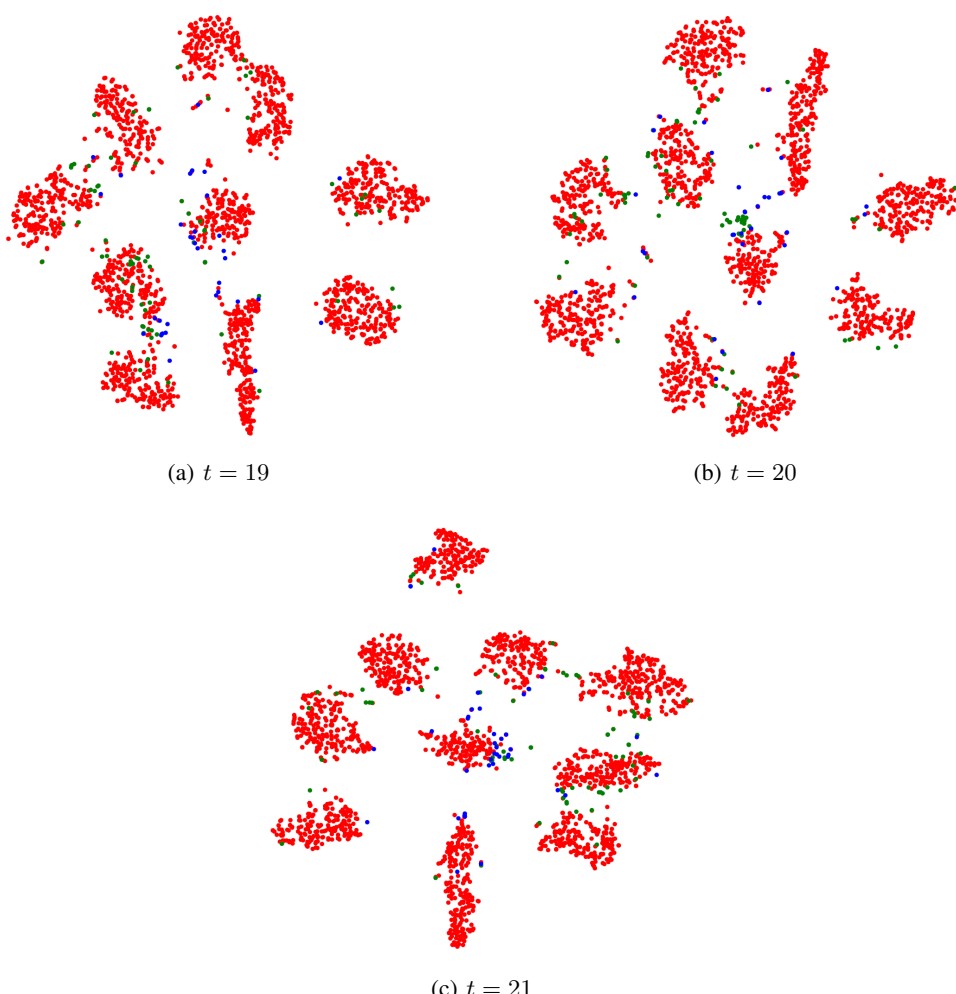

(a) $t = 19$

(b) $t = 20$

(c) $t = 21$

Figure 6: Visualization (IV) of the data points in each domain using t-SNE. We use red points to denote the correctly pseudo-labeled data predicted by the model. We use blue and green points to denote the data with incorrect pseudo-labels, where the blue points are successfully filtered by the filter and the green points are retained. Best viewed in color.