# OpenReview forum: "Adversarial Self-Training Improves Robustness and Generalization for Gradual Domain Adaptation"
_NeurIPS.cc/2023/Conference — NeurIPS 2023 poster_

### Official Review · Reviewer_oh7v · 2023-06-25

**Soundness:** 4 excellent
**Presentation:** 4 excellent
**Contribution:** 3 good
**Rating:** 6
**Confidence:** 4

**Summary:**

This paper investigates the adversarial gradual domain adaptation. The authors apply adversarial learning in the self-training approaches in gradual domain adaptation, forming the adversarial self-training approach (AST). Empirically, the authors show that AST improves both adversarial accuracy and clean accuracy on MNIST and Portraits, and further experimentally explore the reason for the improvements in clean accuracy. Theoretically, new bounds of the empirical risk based on Margin Disparity Discrepancy are developed, providing a tighter bound for adversarial self-training than standard self-training and partially explaining the superiority of AST over standard self-training for gradual domain adaptation.

**Strengths:**

It is appealing that adversarial self-training can improve clean accuracy, for which the authors provide theoretical and empirical analysis.

The theoretical bounds and analysis are new to me.

The paper is well-written and the method is easy to follow.


**Weaknesses:**

In the Introduction, the contributions could be better summarized. To my understanding, the main contributions are the experimental findings on the performance improvements by AST, and the theoretical bounds and analysis, which should be clearly summarized.

The paper assumes the distribution shifts slightly, which could not be realistic.

MNIST and Portraits seem to be simple. Experiments on more challenging datasets are encouraged.



**Questions:**

Line 46, labeling -> labels.

Why should we prove the two inequalities in Lemma A.1? Is it because $\tilde{R}_{P}^{(\rho)}(f_1,f_2)$ (I abbreviate it as $R12$) is asymmetric? If so, do we need to prove the other two inequalities: $R12 \leq R31 + R32$ and $R12 \leq R31 + R23$.

**Limitations:**

Limitations are encouraged to be included.

---

> ### Author Rebuttal · Authors · 2023-08-07
>
> We thank the reviewer for their constructive comments and positive assessment. We are glad that the empirical improvement and theoretical analysis are appealing to you.
>
> **Q1: In the Introduction, the contributions could be better summarized.**
>
> **A1:** Thank you for this important suggestion. The novelty and contribution of our work are mainly focused on:
> * We are the **first** to apply the AST method in gradual domain adaptation and propose gradual AST to improve performance in the gradual domain adaptation setting. (Section 3)
> * The proposed gradual AST method can not only improve the adversarial robustness **but also the clean accuracy** for gradual domain adaptation, which is an **appealing and nontrivial** result, considering that many prior works demonstrate that adversarial training may hurt generalization. (Section 3.1)
> * We empirically explore **the reason for the improvements in clean accuracy**: We find that adversarial training performs better than standard training when the pseudo-labeled training set contains a proportion of incorrect labels. (Section 3.2)
> * We provide the generalization error bounds for gradual AST, which **explain why the trained model is adversarially robust on the target domain**. (Section 4.1)
> * We provide an error bound and a toy example of Gaussian distribution to theoretically **explain the reason for the improvements in clean accuracy**. (Sections 4.2, 4.3)
>
> Thanks again for raising this concern, and we will include a clear summary of our contributions in the revision.
>
> **Q2: The paper assumes the distribution shifts slightly, which could not be realistic.**
>
> **A2:** Thanks for raising this concern. We want to make a clarification that **we only assume that the shift between two adjacent domains is slight, rather than the shift between the source and the target**. In this paper, we assume the learner is provided with a source domain, multiple intermediate domains, and a target domain, and the distribution shifts gradually along the domain sequence. **This is a mild assumption widely accepted in prior works** [A-C]. And there are **many cases in real-world scenarios that satisfy such an assumption**. For example, the sensor measurements shift gradually over time due to sensor aging [D]; the road condition evolves as the car running [E]; and neural signals received by brain-machine interfaces change within the span of a day [F].
>
> [A]: Ananya Kumar et.al, Understanding Self-Training for Gradual Domain Adaptation. ICML 2020.
>
> [B]: Haoxiang Wang et.al, Understanding Gradual Domain Adaptation: Improved Analysis, Optimal Path and Beyond. ICML 2022.
>
> [C]: Hong-You Chen et.al, Gradual Domain Adaptation without Indexed Intermediate Domains. NeurIPS 2021.
>
> [D]: Alexander Vergara et.al, Chemical gas sensor drift compensation using classifier ensembles. Sensors and Actuators B: Chemical.
>
> [E]: Andreea Bobu et.al, Adapting to Continuously Shifting Domains. ICLR 2018.
>
> [F]: Ali Farshchian et.al, Adversarial Domain Adaptation for Stable Brain-Machine Interfaces. ICLR 2019.
>
> **Q3: Experiments on more challenging datasets are encouraged.**
>
> **A3:** Thank you for this constructive suggestion. We train the model on CIFAR-10 and CIFAR-100 datasets using ResNet-50 as the backbone.  From Table 7, we can see that **the gradual AST method consistently improves the clean accuracy and the adversarial accuracy on CIFAR-10 and CIFAR-100 datasets**. Hence, the effectiveness of the gradual AST method is also validated on the larger datasets. We will include these results in the revision.
>
> **Q4: Line 46, labeling -> labels.**
>
> **A4:** We have fixed the typos. Thanks for catching them.
>
> **Q5: Why should we prove the two inequalities in Lemma A.1? Is it because $R_{12}$ is asymmetric? If so, do we need to prove the other two inequalities: $R_{12}\le R_{31}+R_{32}$ and $R_{12}\le R_{31}+R_{23}$?**
>
> **A5:** We prove Lemma A.1 in order to be able to decompose the error in Lemma A.4 (in Eq. (9)). Actually, we just need to use the second inequality in Lemma A.1 and we can remove the first inequality. As for the other two inequalities: $R_{12}\le R_{31}+R_{32}$ and $R_{12}\le R_{31}+R_{23}$, **we don't need to prove them, because the proof of our theory is not based on these two inequalities**.

---

> > ### Comment · Reviewer_oh7v · 2023-08-16
> >
> > I thank the authors for the response. My major concern has been addressed and I maintain the score.

---

> > > ### Author Response · Authors · 2023-08-16
> > > **Thanks for your reply!**
> > >
> > > Dear Reviewer oh7v,
> > >
> > > We are glad that our responses address your concern. Thanks again for your positive assessment!

---

### Official Review · Reviewer_z2iW · 2023-06-29

**Soundness:** 2 fair
**Presentation:** 3 good
**Contribution:** 2 fair
**Rating:** 5
**Confidence:** 3

**Summary:**

The paper proposes an Adversarial Self-Training method to improve the robustness and clean accuracy under the scenario of gradual domain adaptation. More specifically, the paper uses networks to generate pesudo label for gradually changing domain and update the model with the pesudo label. Experiments on Rotating MNIST and Portraits show that the proposed paper large improves the robustness of the network. Additionally, it can also improves the clean accuracy. The paper also perform theoretical analysis of their method, providing error bound of AST and provide a toy example demonstrating why AST achieves better clean performance.

**Strengths:**

1. The paper proposes a simple but effective method to achieve robustness under the scenario of gradual domain adaptation.
2. Both theoretical and empirical results are present to show the effectiveness of AST. Particularly, the theoretical analysis in Section 4.1 is clear and useful.
3. Generally, the paper is easy to follow with clear motivation and good presentation.

**Weaknesses:**

1. The proposed method, AST, seems to be an application of [1, 2] under the scenario of gradual domain adaptation, which lacks of novelty.
2. The experiment is also weak. MNIST is too small and simple to prove the effectiveness of AST in practical applications. I will recommend the authors to perform more experiments in the following aspects:

    1). Perform one experiments on gradual domain shift for natural images like CIFAR10.

    2). Enlarging $\epsilon$ to see whether clean accuracy is still improved for larger adversarial budget.
3. Theorem 4.9 seems to imply strong assumption that only if adversarial risk is well controlled, then standard training is dominated by adversarial training. But the paper does not give how well should the adversarial risk gets controlled.

[1] Yair Carmon et.al., Unlabeled data improves adversarial robustness, NeurIPS 2019

[2] Jonathan Uesato et.al., Are Labels Required for Improving Adversarial Robustness?, NeurIPS 2019

**Questions:**

1. Can the authors give high level intuition of Definition 4.3? What does Adversarial Margin Hypothesis Class mean?

**Limitations:**

See Weakness.

---

> ### Author Rebuttal · Authors · 2023-08-07
>
> Thank you for your constructive feedback on our work.
>
> **Q1: The proposed method, AST, seems to be an application of [1,2] under the scenario of gradual domain adaptation (GDA), which lacks of novelty.**
>
> **A1:** Thank you for raising the concern. As you mentioned, some AST similar methods were previously used in [1,2] to improve the adversarial robustness. However, our work is the **first** to apply the AST in GDA and propose gradual AST to improve the performance in the GDA setting. **And most importantly**, we find that gradual AST can not only improve the adversarial robustness but also the **clean accuracy. Other reviewers also agree that this is an appealing and nontrivial result**, considering many prior works demonstrate that adversarial training (AT) may hurt generalization. Novelly, to explore the reason for the improvements in clean accuracy, we empirically find that **AT performs better than standard training when the pseudo-labeled training set contains a proportion of incorrect labels**. From the theoretical perspective, we provide the **generalization error bounds for gradual AST**, which explain why the model is adversarially robust on the target domain. Our results novelly extend the error bounds into the setting of adversarial GDA and multiclass classification. And we provide **an error bound and a toy example of Gaussian distribution** to theoretically explain the reason for the improvements in clean accuracy.
>
> Our work has different focuses on the AST method: the work of [1, 2] focuses on the importance of unlabeled data and shows that the unlabeled data can be incorporated into the training process by the AST method to improve the robustness; while in our work, we point out the gradual AST performs better than the vanilla gradual self-training and the reason of the superiority.
>
> To sum up, the proposed method is only a small part of our contribution, but the surprising and appealing performance improvement (especially for the clean accuracy) and the theoretical and empirical exploration of the reason for the improvements constitute the main contribution of the paper.
>
> **Q2: I will recommend the authors to perform more experiments in the following aspects: ...**
>
> **A2:** Thanks for your important suggestion. We have conducted more experiments for datasets CIFAR10 and CIFAR100 and various perturbation values $\epsilon$. Please refer to the Common Response.
>
> **Q3: Theorem 4.9 seems to imply strong assumption that only if adversarial risk is well controlled, then standard training is dominated by adversarial training. But the paper does not give how well should the adversarial risk gets controlled.**
>
> **A3:** Thank you for raising this excellent concern. To answer this question, we need to take a closer look at the upper bound of Theorem 4.9:
> $\widetilde{R} _{P _\eta}(h)+\Psi^*(p,p _\eta,e,\epsilon).$
> For standard training, where $\epsilon=0$, the upper bound is:
> $R _{P _\eta}(h)+\Psi^*(p,p _\eta,e,0).$ Comparing these two upper bounds, we can conclude that if $\widetilde{R} _{P _\eta}(h)-R _{P _\eta}(h)\le\Psi^*(p,p _\eta,e,0)-\Psi^*(p,p _\eta,e,\epsilon)$, then AT yields a tighter generalization bound than standard training. The term $\widetilde{R} _{P _\eta}(h)-R _{P _\eta}(h)$ is referred to as "smoothness loss" in some works of AT [2,3], which reflects the smoothness of the model $h$. And AT has an implicit smoothness regularization [2,3] which reduces the smoothness loss. If the smoothness loss is well controlled by AT to be less than $\Psi^*(p,p _\eta,e,0)-\Psi^*(p,p _\eta,e,\epsilon)$, then AT may improve the clean accuracy as we observed in the experiments.
>
> In our toy example presented in Section 4.3, the smoothness loss is well controlled by AT. When $w<-\epsilon$, the smoothness loss of $h _{adv}$ equals $0$, i.e., $\widetilde{R} _{P _\eta}(h _{adv})=R _{P _\eta}(h _{adv})$. And $\Psi^*(p,p _\eta,e,0)-\Psi^*(p,p _\eta,e,\epsilon)$ is positive. In this case, AT yields a tighter generalization guarantee than standard training.
>
> **Q4: Can the authors give high level intuition of Definition 4.3? What does Adversarial Margin Hypothesis Class mean?**
>
> **A4:** Thanks for raising the concern. Given a fixed pair of score functions $f,f'$, the composed function in the class $\widetilde{\rho_\mathcal{F}\mathcal{F}}$ reflects how well $f'$ fits $f$ under the adversarial perturbation.
>
> We can analyze this complex hypothesis class step by step. In a standard learning scenario, supervised multi-class classification problem, classical generalization bounds are based on the Rademacher complexity of a composed hypothesis class, which is composed by the loss function and the predictor class. Given a score-based predictor class $\mathcal{F}\triangleq\\{f:\mathcal{X}\to R^{|\mathcal{Y}|}\\}$, and a margin loss $\rho _f(x,y)\triangleq f _y(x)-\underset{y'\neq y}{max}f _{y'}(x)$, then the composed hypothesis class can be defined as
> $$\rho _\mathcal{F}\triangleq\\{(x,y)\to \rho _{f}(x,y):f\in\mathcal{F}\\}=\\{(x,y)\to f _y(x)-\underset{y'\neq y}{max}f _{y'}(x):f\in\mathcal{F}\\}.$$
>
> For GDA, **the label function is not fixed**: during the training process, the pseudo-labels on the current domain are predicted by the model trained on the last domain. Hence, we adapt the composed hypothesis class with an unfixed label function:
> $$\rho _\mathcal{F}\mathcal{F}\triangleq\\{x\to \rho _{f'}(x,h _f(x)):f,f'\in\mathcal{F}\\}.$$
>
> Finally, we add a perturbation to the input $x$ and we have the Adversarial Margin Hypothesis Class:
> $$\widetilde{\rho _\mathcal{F}\mathcal{F}}\triangleq\\{x\to \max _{\|\delta\| _q\le\epsilon}\rho _{f'}(x+\delta,h _f(x)):f,f'\in\mathcal{F}\\}.$$
>
> **References:**
>
> [1] Yair Carmon et.al., Unlabeled data improves adversarial robustness, NeurIPS 2019.
>
> [2] Jonathan Uesato et.al., Are Labels Required for Improving Adversarial Robustness?, NeurIPS 2019.
>
> [3]: Hongyang Zhang et.al, Theoretically Principled Trade-off between Robustness and Accuracy. ICML 2019.

---

> > ### Comment · Reviewer_z2iW · 2023-08-19
> > **Post-rebuttal comments**
> >
> > Thanks for the valuable response from the reviewers. I am impressed by the good performance on CIFAR10, and I agree that gradual AST is different from [1, 2] so that the paper does make some contributions.
> >
> > In terms of Theorem 4.9, I think directly comparing two upper bounds is not appropriate as it does $a<b, c<d, b<d$ cannot imply $a<c$.
> >
> > Based on the good empirical performance, I will increase my score to 5. But I still doubt the whether Theorem 4.9 can explain the improvement on clean performance.

---

> > > ### Author Response · Authors · 2023-08-20
> > >
> > > Dear Reviewer z2iW,
> > >
> > > Thanks for your reply! We are glad that our responses address your concerns about the novelty and contribution of our work.
> > >
> > > As for Theorem 4.9, we want to clarify that our theorem is aimed at providing some insights to the research field about the unexpected improvement of AST. Theorem 4.9 is based on a very general setting without specifying the hypothesis class, the algorithm, or the data distribution. It is usually a very difficult problem to directly prove $a<c$ in such a general setting ($a,c$ are the symbols used in your last reply). Actually, the values of $a$ and $c$ in our setting depend on the complicated optimization process. In many theoretical works, however, researchers investigate the problems by analyzing their upper bounds. Some works [1,2,3] consider that the upper bound can reflect the original quantity to some extent. For example, in the work of [1], the authors prove that the adversarial Rademacher complexity is larger than the standard Rademacher complexity. And as the generalization bound [1] shows, the Rademacher complexity is a key quantity in the upper bound of the generalization gap. Thus, the authors of [1] claim that this could explain why generalization is harder in the adversarial setting by comparing the two upper bounds. Tighter upper bounds are also widely used as optimization functions. For example, the works of [2,3] derive tighter bounds than prior works and propose more powerful methods based on the tighter bound. In conclusion, we think the comparison of the upper bounds in Theorem 4.9 can provide some intuitions for our empirical finding.
> > >
> > > Finally, we also provide a straight case of the Gaussian model in Section 4.3, where we prove that adversarial training can indeed improve clean accuracy in this setting. To our knowledge, the Gaussian model is a commonly used setting [4] in the theoretical works of machine learning.
> > >
> > > Thanks again for your time and efforts! We will be happy to answer any further questions you may have.
> > >
> > > [1]: Dong Yin et.al, Rademacher Complexity for Adversarially Robust Generalization.
> > >
> > > [2]: Shai Ben-David et.al, A theory of learning from different domains.
> > >
> > > [3]: Yaroslav Ganin et.al, Domain-Adversarial Training of Neural Networks.
> > >
> > > [4]: Ludwig Schmidt et.al, Adversarially Robust Generalization Requires More Data.

---

> ### Author Response · Authors · 2023-08-18
>
> Dear Reviewer z2iW,
>
> Thank you again for your detailed and constructive review. We provided more experiments in our last response on CIFAR datasets and larger adversarial budgets, which consistently verify the effectiveness of our method. We also made more clear clarifications about the novelty and contributions of our work.
>
> Since there are only a few days left in the discussion period, we sincerely ask if we have solved your problem, and we would appreciate the opportunity to engage further if needed. Thanks a lot for your time and efforts!

---

> ### Comment · Area_Chair_XwTQ · 2023-08-18
> **Please provide feedback to the authors' rebuttal**
>
> Dear Reviewer z2iw,
>
> As the end of the discussion period draws near, I kindly remind you about providing feedback on the authors' rebuttal. Your insights are valuable in ensuring a comprehensive review process. Please consider reviewing the feedback from other reviewers as well before formulating your response.
>
> Your timely engagement is greatly appreciated.
>
> Best regards, AC

---

### Official Review · Reviewer_QQok · 2023-07-03

**Soundness:** 3 good
**Presentation:** 4 excellent
**Contribution:** 3 good
**Rating:** 6
**Confidence:** 3

**Summary:**

This paper proposes adversarial self-training (AST) that incorporates adversarial training with gradual self-training. Empirical results validate that AST can enhance both adversarial robustness and generalization ability in the scenario of the gradual unsupervised domain adaption. The authors provide empirical and theoretical analyses to interpret this surprising phenomenon.

**Strengths:**

1. This paper is well-organized and well-written.

2. The empirical observation, that AST can improve robust and natural test accuracy simultaneously in gradual domain adaptation, is interesting and surprising. The natural test accuracy achieved by AST is even close to the optimal results, which validates the significance of the empirical results.

3. The authors provide comprehensive empirical analyses to explain the phenomenon that AST improves natural accuracy.

4. The authors present comprehensive theoretical analyses of AST, including the error bound of (gradual) AST and the connection between natural risk and adversarial risk. The theoretical analyses further help to explain the effectiveness of AST.


**Weaknesses:**

1. The experiments are conducted on a simple network. What if the results were evaluated on a more complex network say ResNet18?

2. I am a bit confused about the term “gradual shifts”. Does this term mean that the test distribution is different from all the training distributions? If so, I am interested in the performance evaluated on the previous training distributions achieved by AST as well to check whether there is a forgetting issue.


**Questions:**


1. Please refer to the comments in the Weaknesses.

2. What kind of adversarial attacks did you use when evaluating adversarial robustness?

3. How can you choose the perturbation bound during AST? What is the performance if using different perturbation bounds?


**Limitations:**

The authors did not discuss the limitations.

---

> ### Author Rebuttal · Authors · 2023-08-07
>
> We thank the reviewer for their constructive comments and positive assessment. We are glad that you find the empirical observation interesting and surprising.
>
> **Q1: What if the results were evaluated on a more complex network say ResNet18?**
>
> **A1:** Thank you for raising the concern. As per your suggestion, we have included additional experiments on ResNet18 and ResNet50. The results can be found in the pdf file in the common response. As Tables 2 and 4 show, **the proposed gradual AST method consistently improves the clean accuracy and adversarial accuracy on large ResNet models**. For example, the clean accuracy on portraits using ResNet18 is improved by $0.52\\%$. And the accuracy achieved by ResNet is higher than that of a 3-layer convolutional neural network. We will include these positive results in the revision paper.
>
> **Q2: Does the term "gradual shifts" mean that the test distribution is different from all the training distributions? If so, I am interested in the performance evaluated on the previous training distributions achieved by AST as well to check whether there is a forgetting issue.**
>
> **A2:** Yes, the term "gradual shifts" means that the distributions of the domains change gradually along the domain sequence and the shift between two adjacent domains is slight. Hence the test distribution is different from the training distributions. As the algorithm of the gradual AST method shows (sections 2.2 and 3), the model is first pre-trained on the source domain and then successively self-trained on the intermediate domains. We record the accuracy of our method on the source domain during the training process. As Fig. 1 shows, when the model is finetuned along the domain sequence, the accuracy on the source domain gradually decreases. **This forgetting phenomenon is foreseeable considering the domain shift and widely studied in the field of continue learning** [A-D]. However, in the (gradual) domain adaptation setting, researchers [E,F] mainly focus on the performance on the target domain where the model is expected to be applied, rather than the performance on the source domain. **We believe the forgetting issue is a matter of concern in some scenarios, but not the focus of the (gradual) domain adaptation setting [E,F].**
>
> [A]: Gido M. van de Ven et.al, Three scenarios for continual learning. arxiv:1904.07734.
>
> [B]: Pietro Buzzega et.al, Dark Experience for General Continual Learning: a Strong, Simple Baseline. NeurIPS 2020.
>
> [C]: Ronald Kemker et.al, Measuring Catastrophic Forgetting in Neural Networks. AAAI 2018.
>
> [D]: Sang-Woo Lee et.al, Overcoming Catastrophic Forgetting by Incremental Moment Matching. NeurIPS 2017.
>
> [E]: Garrett Wilson et.al, A Survey of Unsupervised Deep Domain Adaptation. ACM Transactions on Intelligent Systems and Technology.
>
> [F]: Mingsheng Long et.al, Conditional Adversarial Domain Adaptation. NeurIPS 2018.
>
> **Q3: What kind of adversarial attacks did you use when evaluating adversarial robustness?**
>
> **A3:** We use the **PGD-20** [G] attack to evaluate the adversarial robustness. PGD-20 attack is widely used in many works [G][H] of adversarial training.
>
> **Q4: How can you choose the perturbation bound during AST? What is the performance if using different perturbation bounds?**
>
> **A4:** Following the work of [H], we choose $\epsilon=0.1$ and $\epsilon=0.031$ in the main body for MNIST and portraits respectively. To validate the effectiveness of the gradual AST method using different perturbation bounds, we conduct experiments on MNIST using $\epsilon=0.15, 0.2, 0.3$ and conduct experiments on portraits using $\epsilon=0.01,0.02,0.04,0.05$. As Tables 5 and 6 show, **the gradual AST method can consistently improve the clean accuracy and adversarial robustness using various $\epsilon$ values**.
>
> [G]: Aleksander Madry et.al, Towards Deep Learning Models Resistant to Adversarial Attacks. ICLR 2018.
>
> [H]: Hongyang Zhang et.al, Theoretically principled trade-off between robustness and accuracy. ICML 2019.

---

> > ### Comment · Reviewer_QQok · 2023-08-18
> >
> > Thanks to the authors for the replies. Most of my concerns have been solved. I will keep my score as 6.

---

> > > ### Author Response · Authors · 2023-08-18
> > > **Thanks for your reply**
> > >
> > > Dear Reviewer QQok,
> > >
> > > We are glad that our responses address your concern. Thanks again for your constructive comments and the positive assessment!

---

### Official Review · Reviewer_gQtX · 2023-07-09

**Soundness:** 3 good
**Presentation:** 3 good
**Contribution:** 3 good
**Rating:** 6
**Confidence:** 4

**Summary:**

In this paper, the authors analyze the adversarial robustness of the gradual self-training method with adversarial self-training (AST). AST first predicts labels on the unlabeled data and then adversarially trains the model on the pseudo-labeled distribution. The authors introduce an adversarial self-training for gradual unsupervised domain adaptation in a multi-class problem. They show that when there is a little domain shift between consecutive intermediate domains, their adversarial self-training works better in performance and generality (more robust in the case of missed or bad pseudo-labeled) in comparison to vanilla gradual self-training.
They present the generalization error bounds for gradual AST in a multiclass classification setting and then use the optimal value of the Subset Sum Problem to bridge the standard error on a real distribution and the adversarial error on a pseudo-labeled distribution. Results indicate that AT may obtain a tighter bound than standard training on data with incorrect pseudo-labels. We further present an example of a conditional Gaussian distribution to provide more insights into why gradual AST can improve the clean accuracy for GDA.



**Strengths:**

+ The paper is clearly written and organized, and the key concepts and experimental results are described with enough details to understand. There is a clear discussion on the shortcomings and risks of unsupervised gradual domain adaptation on multi-target classification problems.
+ The authors tackle an important and challenging problem. They propose using adversarial self-training for pseudo-labeling in the context of gradual domain adaptation. In particular, the authors show analytically that ℓq adversarial perturbation for sampling in unlabeled intermediate domains has lower error bounds compared to standard self-training.
+ The supplementary material provides many additional proofs and experimental results that help support the paper.  Also, since the codes are also provided in supplementary material, there is less concern that the results in this paper would be difficult for a reader to reproduce.  The paper includes some information that would make it possible to reproduce the methods and experiments.

**Weaknesses:**

- When comparing to other methods in the literature, the authors claim that their method is based on milder assumptions. They should provide a more detailed discussion of the implications of such assumptions.
- The experimental validation is limited in some respects. Authors show that gradual AST improves both the clean accuracy and adversarial accuracy of the GDA model. Validation is mostly performed on Rotating MNIST and Portraits datasets. However, experiments should assess in more detail the impact on the performance of growing the number of intermediate domains, using different backbones, and different larger datasets with more classes and class imbalance.
- There should be an analysis of time and memory complexity for the proposed and SOA methods.

**Questions:**

None. Please comment on the points described in the weaknesses.

**Limitations:**

Lack of comparison with other SOA domain adaptation methods.  Refer to the weaknesses for details.

---

> ### Author Rebuttal · Authors · 2023-08-07
>
> We thank the reviewer for their detailed and constructive comments.
>
> **Q1: They should provide a more detailed discussion of the implications of the milder assumptions.**
>
> **A1:** Thank you for your constructive suggestion. In the work of [A], the authors assume that the new model can always fit the pseudo-labels without error, which can greatly simplify their analysis. On the contrary, we don't need this strong assumption in our work (Maybe it is better to say "we don't need this assumption" instead of "our results are based on milder assumptions"). **Our assumption is more consistent with real experiments.** The assumption in [A] means that the training error (fitting error) on each intermediate domain should equal to $0$, which does not correspond to the real situation. As we can observe from the experiments, the training error tends to be small but never equal to $0$. And the training error depends on the complexity of the hypothesis class. We will include this more detailed discussion in the revision.
>
> **Q2: The impact on the performance of growing the number of intermediate domains, using different backbones, and different larger datasets with more classes and class imbalance.**
>
> **A2:** Thank you for raising the concern. We have conducted more detailed experiments with different numbers of intermediate domains, different backbones, and different datasets as you suggested (the results can be found in the pdf file in the common response).
> * Larger number of intermediate domains: We split the Rotating MNIST and the portraits datasets with different lengths of interval and conduct an ablation study. For MNIST, we set the numbers of intermediate domains to be 24, 30, and 42 respectively; for portraits, we set the numbers of intermediate domains to be 8, 10, and 14 respectively. As we can observe in Tables 1 and 3, **the gradual AST method consistently improves performance compared to the vanilla gradual self-training method.** The clean accuracy of gradual AST on MNIST falls into a small range of $97.0\\%$ to $97.5\\%$, regardless of the choice of the number of intermediate domains. The results are not sensitive to the number of intermediate domains.
> * Different backbones: We use ResNet-18 and ResNet-50 as the backbones. The results in Tables 2 and 4 show that **with larger ResNet models, the clean and adversarial accuracy are consistently improved**, which shows the effectiveness of our gradual AST method using various backbones. For example, the clean and adversarial accuracy on portraits using ResNet18 is improved by $0.52\\%$ and $52.08\\%$ respectively.
> * Different datasets: We train the model on CIFAR-10 and CIFAR-100 datasets using ResNet-50 as the backbone. Similar to Rotating MNIST, we generate the dataset by rotating each CIFAR image at an angle between $0$ and $10$ degrees. The $50,000$ training set images are divided into three parts: a source domain of $12,000$ images, 3 intermediate domains of $36,000$ images, and a set of validation set. From Table 7, we can see that the gradual AST method improves the clean accuracy and the adversarial accuracy on CIFAR-10 by $1.30\\%$ and $33.85\\%$ respectively. **The effectiveness of the gradual AST method is also validated on larger datasets.**
>
> **Q3: There should be an analysis of time and memory complexity for the proposed and SOA methods.**
>
> **A3:** Thank you for raising the concern. The complexity depends on several factors, such as the size of the dataset $n$, the training epoch number $p$, and the number of iterations used in the PGD attack $k$.
> * Time Complexity: There are two main parts in the training process: predicting pseudo-labels and adapting (retraining) the model. **For the first part, both of these two methods have the same complexity $\mathcal{O}(n)$.** For the second part, the gradual AST method is more time-consuming compared to the vanilla gradual self-training method, since the gradual AST method involves additional adversarial example generation. The model needs to be forward-passed and backpropagated $k$ times to generate adversarial examples in each epoch. **In this period, the time complexities of the gradual AST method and vanilla gradual self-training method are $\mathcal{O}((k+1)pn)$ and $\mathcal{O}(pn)$ respectively.** The complexities between these two methods are of the same order, just one constant time different. And the benefits of greatly improved performance can justify the extra computational cost. Experimentally, for our experiments on an Nvidia GeForce RTX 2080 GPU, **the gradual AST method ($\tau=0$) takes less than one hour to train a model on Rotating MNIST.**
> * Memory Complexity: During the training process of the neural network, the computation graph [C] occupies the major portion of memory. **Both the gradual AST method and vanilla gradual self-training method have the same size of computation graph. The additional memory consumption for storing the adversarial perturbations is almost negligible.** For our experiments on Rotating MNIST with a 3-layer convolutional neural network, the memory consumption for both methods are only 920 MB, which is an affordable cost for most GPUs.
>
>
> **References:**
>
> [A]: Haoxiang Wang et.al, Understanding gradual domain adaptation: Improved analysis, optimal path and beyond. ICML, 2022.
>
> [B]: Shai Ben-David et.al, Understanding machine learning: From theory to algorithms.
>
> [C]: How Computational Graphs are Constructed in PyTorch. https://pytorch.org/blog/computational-graphs-constructed-in-pytorch/

---

> > ### Comment · Reviewer_gQtX · 2023-08-18
> > **Comments on rebuttal.**
> >
> > I thank the authors for their response and the new results. My concerns have been addressed by the author’s response. Based on the rebuttal and comments from other reviewers I raise my rating to Weak Accept.

---

> > > ### Author Response · Authors · 2023-08-18
> > > **Thanks a lot for your time and efforts!**
> > >
> > > Dear Reviewer gQtX,
> > >
> > > Thank you very much! We are glad that our response addressed your concerns.
> > >
> > > Thanks again for the constructive comments and the raised score!

---

### Author Rebuttal · Authors · 2023-08-07

## Common Response (Additional Experiments)
We thank all the reviewers for their detailed and constructive comments. As per suggestions from the reviewers, we include some additional experiments here. **The results can be found in the attached PDF file**.

* **Larger number of intermediate domains**: We split the Rotating MNIST dataset with different lengths of interval and conduct an ablation study. For MNIST, the numbers of intermediate domains are 24, 30, and 42, respectively; for portraits, the numbers of intermediate domains are 8, 10, and 14, respectively. The results can be found in Tables 1 and 3, which show that our gradual AST method is nonsensitive to the choice of the number of intermediate domains.
* **Different backbones**: We use ResNet-18 and ResNet-50 as the backbones. The results can be found in Tables 2 and 4, which validate that the gradual AST method consistently improves performance on large models compared to the vanilla self-training method. For example, the clean and adversarial accuracy on portraits using ResNet18 is improved by $0.52\\%$ and $52.08\\%$ respectively.
* **Different datasets**: We train the model on the CIFAR-10 and CIFAR-100 datasets using ResNet-50 as the backbone. Similar to Rotating MNIST, we generate the dataset by rotating each CIFAR image at an angle between $0$ and $10$ degrees. The $50,000$ training set images are divided into three parts: a source domain of $12,000$ images, 3 intermediate domains of $36,000$ images, and a set of validation set. From Table 7, we can see that the gradual AST method improves the clean accuracy and the adversarial accuracy on CIFAR-10 by $1.30\\%$ and $33.85\\%$ respectively. Hence, the gradual AST method can also improve performance on larger datasets.
* **Different perturbation bound $\epsilon$**: To validate the effectiveness of the gradual AST method using different perturbation bounds, we conduct experiments on MNIST using $\epsilon=0.15, 0.2, 0.3$ and on portraits using $\epsilon=0.01,0.02,0.04,0.05$. The gradual AST method can consistently improve clean accuracy and adversarial robustness using various $\epsilon$ values. Please refer to Tables 5 and 6.

---

### Decision · Program_Chairs · 2023-09-21

**Decision:**

Accept (poster)

**Comment:**

In this paper, the authors propose using adversarial self-training (AST) in gradual self-training for domain adaptation and show that it improves not only adversarial accuracy but also clean accuracy on the target domain in an empirical analysis. The authors reveal that this is because the proposed AST is more robust when the pseudo-labels obtained on the new domain contain a portion of incorrect labels. They further present the generalization error bounds for gradual AST.

The paper received unanimously positive reviews. The reviewers, in general, found the tackled problem of gradual domain adaptation to be important, and the empirical finding that AST can improve both robust and clean accuracy in such a setting to be interesting and a meaningful contribution. They also found the explanation of this phenomenon and the theoretical analysis to be comprehensive and useful, and the paper to be well-written.

However, there were concerns regarding the results obtained with smaller networks on simple datasets, a lack of complexity analysis, a lack of novelty, and strong assumptions in the theoretical analysis. Nevertheless, they were less critical, and the authors' responses with additional experimental results cleared away most of the concerns. Thus, the reviewers maintained their positive ratings even after the rebuttal, reaching a consensus to accept the paper.